# DURIAN: DUAL REFERENCE IMAGE-GUIDED PORTRAIT ANIMATION WITH ATTRIBUTE TRANSFER

**Hyunsoo Cha, Byungjun Kim, Hanbyul Joo**
Seoul National University
{243stephen,byungjun.kim,hbjoo}@snu.ac.kr

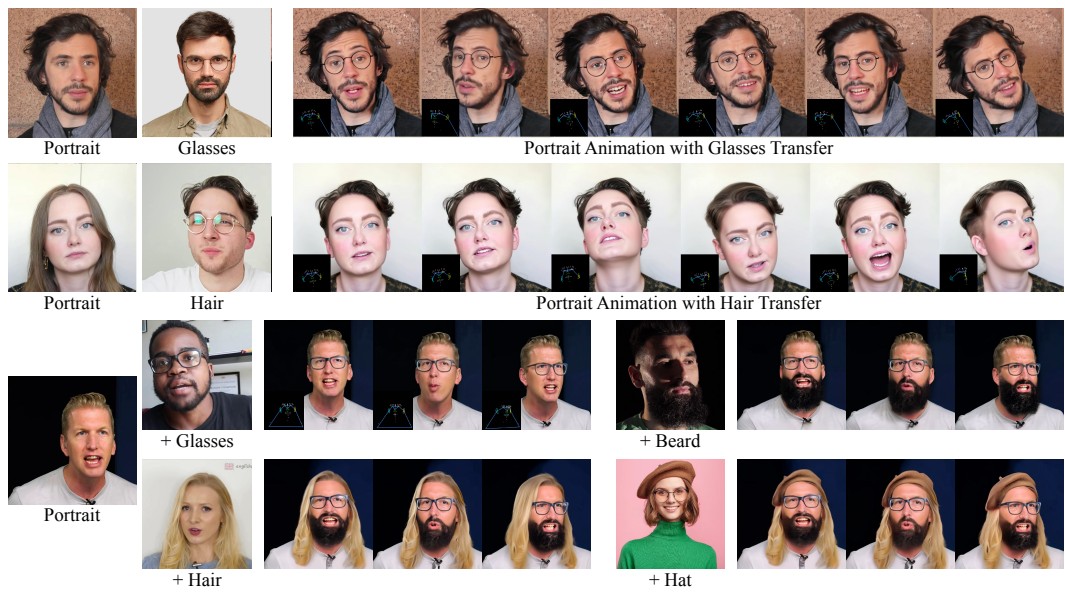

Figure 1: **Portrait Animation with Attribute Transfer.** Given a portrait image and single or multiple reference images specifying target attributes (*e.g.*, hairstyle, eyeglasses), our method generates a portrait animation with facial attribute transfer conditioned on a keypoint sequence.

## ABSTRACT

We present Durian, the first method for generating portrait animation videos with cross-identity attribute transfer from one or more reference images to a target portrait. Training such models typically requires attribute pairs of the same individual, which are rarely available at scale. To address this challenge, we propose a self-reconstruction formulation that leverages ordinary portrait videos to learn attribute transfer without explicit paired data. Two frames from the same video act as a pseudo pair: one serves as an attribute reference and the other as an identity reference. To enable this self-reconstruction training, we introduce a Dual ReferenceNet that processes the two references separately and then fuses their features via spatial attention within a diffusion model. To make sure each reference functions as a specialized stream for either identity or attribute information, we apply complementary masking to the reference images. Together, these two components guide the model to reconstruct the original video, naturally learning cross-identity attribute transfer. To bridge the gap between self-reconstruction training and cross-identity inference, we introduce a mask expansion strategy and augmentation schemes, enabling robust transfer of attributes with varying spatial extent and misalignment. Durian achieves state-of-the-art performance on portrait animation with attribute transfer. Moreover, its dual reference design uniquely supports multi-attribute composition and smooth attribute interpolation within a single generation pass, enabling highly flexible and controllable synthesis. Project page and code are available at: https://hyunsoocha.github.io/durian

# 1 INTRODUCTION

Personalized appearance editing, such as virtually trying on glasses or experimenting with new hairstyles, is becoming a key feature of virtual styling applications. However, most existing solutions are highly specialized and limited in scope. Hairstyle preview apps typically rely on fixed templates, which may look realistic from a single view but fail to adapt to head pose or expression changes. Glasses try-on systems often depend on pre-scanned 3D product models, restricting users to a predefined catalog. Furthermore, these systems focus on a single attribute and cannot combine multiple elements, such as hair, glasses, or hats, within a unified experience.

A key challenge in building such a system is obtaining suitable training data. Disentangling identity from attributes ideally requires paired images of the same person with different attributes, which are rarely available and expensive to collect at scale. This difficulty grows exponentially for multiple attributes, as capturing all combinations quickly becomes infeasible. For example, Li et al. (2023) collect multi-view images of subjects wearing different eyeglasses to model realistic glasses try-on, but the dataset remains too limited to generalize broadly. Zhang et al. (2025) propose a synthetic pipeline that predicts a bald version of a portrait and generates reference hair images using a pretrained diffusion model. However, this approach is not easily scalable beyond hair.

This naturally raises the question: *can we train a model for portrait animation with attribute transfer without any explicit attribute-paired data?* Motivated by this question, we propose a **self-reconstruction framework** that learns this task directly from widely available in-the-wild portrait videos. During training, we randomly sample two frames from a single video: one as the attribute reference and the other as the identity reference. The remaining frames are treated as targets to be generated, conditioned on a keypoint sequence representing the motion of the video. To prevent identity leakage, we apply complementary masking to the two reference frames so that the network must disentangle and combine the attribute and identity information to reconstruct the original video.

To enable this framework, we design a **Dual ReferenceNet** architecture that explicitly encodes the attribute and portrait references through two separate branches and fuses their disentangled features for generation via spatial attention. This design enables the network to move beyond simple pose driving, generating keypoint-driven portrait animations that seamlessly combine the attribute from one image with the identity from the other. Surprisingly, although the model is trained with only a single attribute reference at a time, the spatial attention mechanism allows more advanced operations at inference time. Since different attributes (*e.g.*, hair, glasses, beard, hats) occupy distinct spatial regions, their features can be jointly injected without conflict, enabling seamless multi-attribute transfer. Furthermore, by interpolating the features of two attribute references, our model can achieve attribute interpolation, generating smooth transitions between the attributes. These emergent capabilities make our framework especially valuable for real-world styling scenarios, where users may want to explore diverse combinations and gradual transformations of facial attributes.

While self-reconstruction training is effective for learning to separate identity and attributes, it operates within a single video, leading to a domain gap when the model is applied to cross-identity inference, where the attribute and portrait come from different individuals. To mitigate this gap, we introduce a mask expansion strategy and lightweight augmentation schemes. These techniques expose the model to a broader range of attribute configurations during training, enabling robust transfer across spatial and structural variations of the attribute region. These designs form a unified framework capable of robust cross-identity attribute transfer. As a result, our method achieves a versatile system that generates portrait animations with diverse appearance edits in a zero-shot manner.

We summarize the key contributions of our work, as follows:

- We propose the first method to generate keypoint-driven portrait animations with transferred attributes directly from two images, generalized across diverse facial attributes beyond hair.
- We design a Dual ReferenceNet architecture that disentangles attribute and identity through two branches fused via spatial attention, enabling self-reconstruction training directly on uncurated in-the-wild videos without paired data.
- We propose a mask expansion strategy and lightweight augmentations to bridge the domain gap for cross-identity transfer, improving robustness to diverse spatial configurations.
- Our framework exhibits an emergent ability to support multi-attribute composition and interpolation in a single generation pass, without requiring any additional training.

## 2 Related Work

**Face Editing.** Generative models have advanced facial editing from unconditional synthesis to fine-grained manipulation of existing images (Goodfellow et al., 2014; Rezende & Mohamed, 2015; Ho et al., 2020). Latent-space editing with StyleGAN (Karras et al., 2020) and GAN inversion (Zhu et al., 2016; Abdal et al., 2019; Richardson et al., 2021) has been extended to video via latent trajectory modeling (Yao et al., 2021; Tzaban et al., 2022) and 3D-aware editing (Bilecen et al., 2024; Xu et al., 2024). However, such approaches often rely on attribute classifiers or fixed editing controls. Diffusion-based models have introduced more flexible editing through prompt-driven (Brooks et al., 2023) or identity-preserving techniques (Ye et al., 2023; Wang et al., 2024), with extensions to video improving temporal consistency (Ku et al., 2024; Kim et al., 2023). Still, these methods are limited to modifying existing content and cannot generate new motions or expressions.

**Diffusion-based Attribute Transfer.** Diffusion-based attribute transfer methods typically formulate editing as masked inpainting, where reference content is inserted into a target image using explicit masks (Yang et al., 2023; Chen et al., 2024; Mou et al., 2025; Chen et al., 2025; Song et al., 2025). These approaches have been adapted to domain-specific tasks such as hairstyle (Zhang et al., 2025; Chung et al., 2025), clothing (Kim et al., 2024a; Li et al., 2024; Chong et al., 2024), and makeup (Zhang et al., 2024b). While effective for static images, they rely on category labels or mask annotations. Video extensions (Fang et al., 2024; Tu et al., 2025) apply per-frame inpainting with post-hoc smoothing, but predefined masks are hard to specify for deformable facial attributes that vary over time. Recent works have also explored attribute transfer in 3D avatars (Kim et al., 2024b; Nam et al., 2025; Cha et al., 2024; 2025; Wang et al., 2025; Kim et al., 2025), but such approaches often require specialized capture setups or are not easily generalizable to in-the-wild scenarios. In contrast, our model performs attribute transfer and animation jointly in a single forward pass, conditioned only on a pair of reference images and a facial keypoint sequence. This eliminates the need for per-frame masks, text prompts, or category labels, enabling zero-shot transfer of diverse facial attributes.

**Portrait Animation from a Single Image.** Portrait animation aims to generate motion from a static image, typically guided by facial keypoints, audio, or motion trajectories. Early methods rely on GANs with implicit keypoint modeling (Guo et al., 2024; Wang et al., 2021), while recent approaches use diffusion models (Hu, 2024; Zhu et al., 2024; Yang et al., 2025) for improved realism and temporal stability. These methods primarily focus on reenactment and identity preservation. Others incorporate paired motion (Xie et al., 2024) or audio (Yang et al., 2025), but require multi-stage inference or fine-tuning. Our model jointly performs facial attribute transfer and motion generation, producing photorealistic, identity-preserving videos from diverse attribute references and keypoint-driven motion in a single pass.

## 3 Method

### 3.1 Overview: Learning Attribute Transfer from Self-Reconstruction

We propose a diffusion-based generative framework for portrait animation with cross-identity attribute transfer. At a high level, our model generates an $F$-frame animation sequence $\mathbf{V} = \{\mathbf{I}_\tau\}_{\tau=1}^{F}$ as:

$$\mathbf{V} = \text{Durian}(\mathbf{I}_{\text{attr}}, \mathbf{M}_{\text{attr}}, \mathbf{I}_{\text{port}}, \mathbf{M}_{\text{port}}, \mathbf{K}), \tag{1}$$

conditioned on an attribute image $\mathbf{I}_{\text{attr}}$, a portrait image $\mathbf{I}_{\text{port}}$, and a sequence of driving facial keypoint images $\mathbf{K} = \{\boldsymbol{k}_\tau\}_{\tau=1}^{F}$. Each reference image has a binary mask: $\mathbf{M}_{\text{attr}}$ localizes the attribute region (*e.g.*, hair or glasses) in the reference image, while $\mathbf{M}_{\text{port}}$ specifies the candidate region in the portrait where the attribute will be transferred. Using these masks, we construct two masked inputs: the *attribute-only image* $\tilde{\mathbf{I}}_{\text{attr}} = \mathbf{I}_{\text{attr}} \odot \mathbf{M}_{\text{attr}}$, where only the attribute region is preserved, and the *attribute-masked portrait image* $\tilde{\mathbf{I}}_{\text{port}} = \mathbf{I}_{\text{port}} \odot (1 - \mathbf{M}_{\text{port}})$, where the corresponding region is removed. These masked inputs are fed into the **Dual ReferenceNet**, consisting of the *Attribute ReferenceNet (ARNet)* and *Portrait ReferenceNet (PRNet)*, which extract multi-scale spatial features. These features are then injected into a diffusion-based generator, the *Denoising UNet (DNet)*, to synthesize the remaining frames of the video with keypoint guidance $\mathbf{K}$ (Section 3.2).

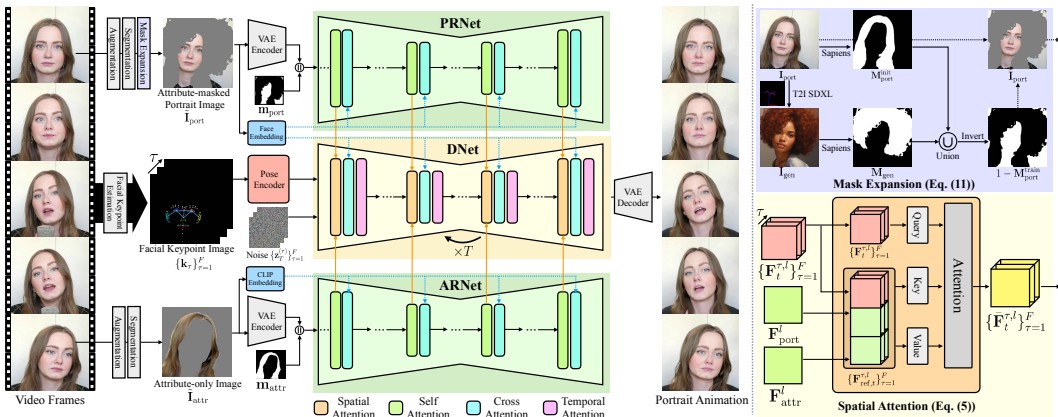

Figure 2: **Overview of Training Pipeline.** Given an attribute-masked portrait image $\tilde{\mathbf{I}}_{\text{port}}$ and an attribute-only image $\tilde{\mathbf{I}}_{\text{attr}}$, Durian synthesizes a portrait animation with the transferred attribute. These inputs are constructed by randomly sampling two frames from a training video and applying the estimated masks. A sequence of facial keypoints $\{\boldsymbol{k}_\tau\}_{\tau=1}^F$ is extracted from the video to guide the motion. During generation, spatial features from PRNet and ARNet are fused via spatial attention into the DNet, ensuring identity preservation and attribute consistency in the synthesized video.

To enable training without requiring explicitly annotated triplets (*i.e.*, combinations of a target attribute image, an original portrait image, and an edited portrait image), we adopt a **self-reconstruction strategy** based on portrait videos (Yu et al., 2023; Xie et al., 2022). Specifically, we simulate attribute transfer by sampling two frames $\mathbf{I}_{\text{attr}}$ and $\mathbf{I}_{\text{port}}$ from the same video, treating one as the attribute reference and the other as the target portrait. We then construct the masked inputs $\tilde{\mathbf{I}}_{\text{attr}}$ and $\tilde{\mathbf{I}}_{\text{port}}$ using the same masking formulation as in inference, based on a segmentation mask of a randomly selected attribute. Although the two frames come from the same identity, the complementary masking enforces a clear separation between identity and attribute inputs, encouraging the model to learn meaningful mappings from these features to output frames without requiring cross-identity supervision. To enhance the model's ability to generalize beyond the self-attribute transfer setup, we introduce an augmentation scheme that improves robustness to spatial and appearance variations(Section 3.3).

At inference time, we estimate refined attribute masks by aligning the attribute image to the portrait through a lightweight alignment process, mitigating spatial misalignment between them. Conditioned on the two masked reference images and the driving keypoint sequence, our model then synthesizes portrait animations with attribute transfer. Notably, our design also supports multi-attribute composition and smooth interpolation within a single generation pass, without requiring additional training or post-processing (Section 3.4). Fig. 1 shows our generated portrait animations with attribute transfer.

## 3.2 MODEL ARCHITECTURE: DUAL REFERENCENET

Inspired by recent approaches (Guo et al., 2023; Hu, 2024; Zhu et al., 2024) that leverage ReferenceNet to inject spatial features into diffusion models, we propose a **Dual ReferenceNet** architecture tailored for portrait animation with attribute transfer. Unlike previous work, our model includes two separate encoders: *Attribute ReferenceNet (ARNet)* and *Portrait ReferenceNet (PRNet)*, each sharing the same architecture as the *Denoising U-Net (DNet)* in the diffusion model, excluding the temporal layers. The networks follow the U-Net (Long et al., 2015) architecture used in latent diffusion models (Rombach et al., 2022), with each block containing convolutional layers followed by self- and cross-attention modules. The overall architecture is shown in Fig. 2.

**Reference inputs.** Given an attribute image $\mathbf{I}_{\text{attr}} \in \mathbb{R}^{3 \times H \times W}$ and a portrait image $\mathbf{I}_{\text{port}} \in \mathbb{R}^{3 \times H \times W}$, along with their binary masks $\mathbf{M}_{\text{attr}} \in \mathbb{R}^{1 \times H \times W}$ and $\mathbf{M}_{\text{port}} \in \mathbb{R}^{1 \times H \times W}$, which localize the attribute region and the candidate transfer region respectively, we construct two masked inputs: the attribute-only image $\tilde{\mathbf{I}}_{\text{attr}} = \mathbf{I}_{\text{attr}} \odot \mathbf{M}_{\text{attr}}$, where only the attribute region is preserved, and the attribute-masked portrait image $\tilde{\mathbf{I}}_{\text{port}} = \mathbf{I}_{\text{port}} \odot (1 - \mathbf{M}_{\text{port}})$, where the corresponding candidate

region is removed. We then encode these masked images into latent representations using the pretrained VAE from the latent diffusion model (Rombach et al., 2022), yielding $\boldsymbol{z}_{\text{attr}}, \boldsymbol{z}_{\text{port}} \in \mathbb{R}^{c \times h \times w}$. The corresponding masks $\mathbf{M}_{\text{attr}}, \mathbf{M}_{\text{port}}$ are downsampled to match the latent resolution, producing $\boldsymbol{m}_{\text{attr}}, \boldsymbol{m}_{\text{port}} \in \mathbb{R}^{1 \times h \times w}$. These downsampled masks are concatenated with the latents along the channel dimension to form $(c+1)$-channel inputs $\tilde{\boldsymbol{z}}_{\text{attr}}, \tilde{\boldsymbol{z}}_{\text{port}} \in \mathbb{R}^{(c+1) \times h \times w}$ as follows:

$$\tilde{\boldsymbol{z}}_{\text{attr}} = \text{concat}_{\text{c}}(\boldsymbol{z}_{\text{attr}}, \boldsymbol{m}_{\text{attr}}), \quad \tilde{\boldsymbol{z}}_{\text{port}} = \text{concat}_{\text{c}}(\boldsymbol{z}_{\text{port}}, \boldsymbol{m}_{\text{port}}). \tag{2}$$

**Spatial attention.** The augmented latents are passed to ARNet $\mathcal{E}_{\text{attr}}$ and PRNet $\mathcal{E}_{\text{port}}$ to extract multi-scale feature maps after convolutional layers of each block:

$$\mathcal{F}_{\text{attr}} \coloneqq \{\mathbf{F}_{\text{attr}}^l\}_{l=1}^L = \mathcal{E}_{\text{attr}}(\tilde{z}_{\text{attr}}; \Theta_{\text{attr}}), \quad \mathcal{F}_{\text{port}} \coloneqq \{\mathbf{F}_{\text{port}}^l\}_{l=1}^L = \mathcal{E}_{\text{port}}(\tilde{z}_{\text{port}}; \Theta_{\text{port}}), \tag{3}$$

where $\Theta_{\{\text{attr},\text{port}\}}$ are the parameters of Dual ReferenceNet. Let $\mathbf{F}_t^{\tau,l} \in \mathbb{R}^{c_l \times h_l \times w_l}$ denote the feature map of the frame $\tau$ at the $l$-th block of the denoising U-Net. While the original denoising U-Net includes a self-attention layer at each resolution, we replace it with our spatial attention to integrate identity and attribute features in a spatially-aware manner. We denote width-wise concatenation as $\text{concat}_{\text{w}}(\cdot)$, and define our spatial attention $\text{SA}(\cdot, \cdot, \cdot)$ as:

$$\mathbf{F}_{\text{ref},t}^{\tau,l} \coloneqq \text{concat}_{\text{w}}(\{\mathbf{F}_t^{\tau,l}, \mathbf{F}_{\text{port}}^l, \mathbf{F}_{\text{attr}}^l\}) \in \mathbb{R}^{c_l \times h_l \times 3w_l}, \tag{4}$$

$$\bar{\mathbf{F}}_t^{\tau,l} = \text{SA}(\mathbf{F}_t^{\tau,l}, \mathbf{F}_{\text{port}}^l, \mathbf{F}_{\text{attr}}^l) = \text{Attention}(\mathbf{W}_Q \mathbf{F}_t^{\tau,l}, \mathbf{W}_K \mathbf{F}_{\text{ref},t}^{\tau,l}, \mathbf{W}_V \mathbf{F}_{\text{ref},t}^{\tau,l}), \tag{5}$$

where $\bar{\mathbf{F}}_t^{\tau,l} \in \mathbb{R}^{c_l \times h_l \times w_l}$ is the feature map after the spatial attention, $\text{Attention}(Q, K, V) = \text{softmax}(QK^\top/\sqrt{d})V$ is the standard scaled dot-product attention (Vaswani et al., 2017), $\mathbf{W}_Q, \mathbf{W}_K, \mathbf{W}_V$ are linear projection layers. This width-wise concatenation preserves spatial resolution and allows the model to attend across all positions in the combined reference and target features. As a result, the model can leverage both attribute and portrait guidance at every step.

**Cross-attention with semantic embeddings.** After applying spatial attention, we further inject semantic guidance into both the Dual ReferenceNet and the denoising U-Net via cross-attention. For ARNet, we use the CLIP (Radford et al., 2021) embedding of the attribute-only image $\tilde{\mathbf{I}}_{\text{attr}}$ as the attribute embedding $\phi_{\text{attr}}$, which is injected via cross-attention into each block of ARNet. For PRNet and DNet, we construct a portrait embedding $\phi_{\text{port}}$ by combining ArcFace (Deng et al., 2019) and CLIP embeddings of the attribute-masked portrait image $\tilde{\mathbf{I}}_{\text{port}}$ following StableAnimator (Tu et al., 2024). This embedding is injected into both PRNet and DNet to enhance identity preservation. We define the cross-attention operation $\text{CA}(\cdot, \cdot)$ as:

$$\text{CA}(\bar{\mathbf{F}}, \phi) = \text{Attention}(\mathbf{W}_Q' \bar{\mathbf{F}}, \mathbf{W}_K' \phi, \mathbf{W}_V' \phi), \tag{6}$$

where $\bar{\mathbf{F}}$ is the input feature map, $\phi$ is the conditioning embedding, and $\mathbf{W}_Q', \mathbf{W}_K', \mathbf{W}_V'$ are learned linear projections. Let $\bar{\mathbf{F}}_{\text{attr}}^l$ and $\bar{\mathbf{F}}_{\text{port}}^l$ be the self-attended features of the $l$-th block in ARNet and PRNet, and $\bar{\mathbf{F}}_t^l$ the spatially attended feature of DNet. Then, the cross-attention updates are given by:

$$\tilde{\mathbf{F}}_{\{\text{attr},\text{port}\}}^l = \text{CA}(\bar{\mathbf{F}}_{\{\text{attr},\text{port}\}}^l, \phi_{\{\text{attr},\text{port}\}}), \quad \tilde{\mathbf{F}}_t^{\tau,l} = \text{CA}(\bar{\mathbf{F}}_t^{\tau,l}, \phi_{\text{port}}), \tag{7}$$

where $\tilde{\mathbf{F}}_{\text{attr}}^l, \tilde{\mathbf{F}}_{\text{port}}^l$, and $\tilde{\mathbf{F}}_t^{\tau,l}$ are the feature maps after cross-attention in ARNet, PRNet, and DNet.

**Temporal extension and keypoint guidance.** Our model incorporates temporal awareness to generate coherent portrait animations by inserting temporal self-attention into each U-Net block, following Hu (2024); Zhu et al. (2024). To control pose and expression, we use a sequence of facial keypoints $\mathbf{K} = \{\boldsymbol{k}_\tau\}_{\tau=1}^F$ extracted by Sapiens (Khirodkar et al., 2024). Each keypoint image $\boldsymbol{k}_\tau$ is encoded into a spatial feature map $\mathbf{F}_{\text{kpt}}^\tau$ via a pose encoder and combined with the noisy latent $\boldsymbol{z}_t^{(\tau)}$ following Zhu et al. (2024). For each frame $\tau$, DNet $\epsilon_\theta$ predicts the added noise $\hat{\epsilon}_t^{(\tau)}$ from the noisy latent $\boldsymbol{z}_t^{(\tau)}$ at timestep $t$, using the reference features, semantic embeddings, and keypoint features:

$$\hat{\epsilon}_t^{(\tau)} = \epsilon_\theta\left(\boldsymbol{z}_t^{(\tau)}, t, \mathcal{F}_{\text{attr}}, \mathcal{F}_{\text{port}}, \phi_{\text{attr}}, \phi_{\text{port}}, \mathbf{F}_{\text{kpt}}^\tau\right). \tag{8}$$

The predicted noise is used to recover the denoised latent $\boldsymbol{z}_0^{(\tau)}$, then decoded by the VAE decoder $\mathcal{D}$ to produce the final video frame as $\mathbf{I}_\tau = \mathcal{D}(\boldsymbol{z}_0^{(\tau)})$ for $\tau = 1, \dots, F$.

## 3.3 TRAINING STRATEGY

**Training loss.** To effectively train our model, we adopt a two-stage training scheme following the previous approaches (Hu, 2024; Zhu et al., 2024). In the first stage, we optimize the entire model except the temporal attention layers, treating each video frame as an independent training sample. We define the per-frame conditioning bundle as $\mathcal{C} := (\mathcal{F}_{\text{attr}}, \mathcal{F}_{\text{port}}, \phi_{\text{attr}}, \phi_{\text{port}})$, where $\mathcal{F}_{\text{port}}, \mathcal{F}_{\text{attr}}$ are the multi-scale spatial features from PRNet and ARNet and $\phi_{\text{port}}, \phi_{\text{attr}}$ are the semantic embeddings. Then, the training objective is the standard denoising diffusion loss:

$$\mathcal{L}_{\text{diff}}^{(1)} = \mathbb{E}_{z_0, \epsilon, t} \left[ \| \epsilon - \epsilon_\theta (z_t, t, \mathcal{C}, \mathbf{F}_{\text{kpt}}) \|^2 \right], \tag{9}$$

where $z_t$ is the noised latent at diffusion timestep $t$, $\epsilon$ is the sampled noise, and $\mathbf{F}_{\text{kpt}}$ is the feature map of the corresponding facial keypoint image. In the second stage, we freeze all modules except the temporal attention layers and train them using multi-frame inputs. The temporal objective considers a sequence of noised latents and corresponding keypoints:

$$\mathcal{L}_{\text{diff}}^{(2)} = \mathbb{E}_{\{z_0^{(\tau)}\}_{\tau=1}^F, \epsilon^{1:F}, t} \left[ \left\| \epsilon^{1:F} - \epsilon_\theta \left( \{z_t^{(\tau)}\}_{\tau=1}^F, t, \mathcal{C}, \{\mathbf{F}_{\text{kpt}}^\tau\}_{\tau=1}^F \right) \right\|^2 \right], \tag{10}$$

where $\epsilon^{1:F} = \{\epsilon^{(\tau)}\}_{\tau=1}^F$ denotes the per-frame noise sequence. This staged training improves convergence and allows the temporal attention module to focus on modeling motion dynamics without disrupting the spatial fidelity learned in the first stage.

**Attribute-aware mask expansion.** To expose the model to diverse spatial extents of facial attributes during training, we introduce an attribute-aware mask expansion strategy, illustrated in the top right of Fig. 2. Given a training frame $\mathbf{I}$, we first select a target attribute (*e.g.*, hair, eyeglasses, beard) and obtain its binary mask $\mathbf{M}_{\text{attr}}$ using Sapiens (Khirodkar et al., 2024). To simulate variation in the shape and coverage of this attribute, we generate a modified image $\mathbf{I}_{\text{gen}}$ with SDXL (Podell et al., 2023) and ControlNet (Zhang et al., 2023), conditioned on the facial keypoints of $\mathbf{I}$ and a text prompt describing an altered appearance (e.g., "long wavy hair").

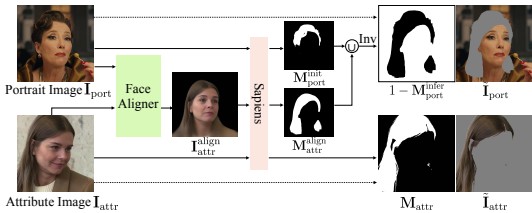

Figure 3: **Aligned Attribute Mask Estimation.** To improve attribute-portrait alignment, we estimate an aligned attribute mask via Face Aligner.

To enable fully automated prompt generation without any human intervention, we construct a dictionary of descriptive attribute modifiers (e.g., long, short, wavy, curly) and randomly sample their combinations to generate prompts for image generation. A new mask $\mathbf{M}_{\text{gen}}$ is then extracted from $\mathbf{I}_{\text{gen}}$ using Sapiens. The final training mask is computed as the union of the original and generated masks, and the two masked inputs are constructed as:

$$\mathbf{M}_{\text{port}}^{\text{train}} = \mathbf{M}_{\text{attr}} \cup \mathbf{M}_{\text{gen}}, \quad \tilde{\mathbf{I}}_{\text{attr}} = \mathbf{I} \odot \mathbf{M}_{\text{attr}}, \quad \tilde{\mathbf{I}}_{\text{port}} = \mathbf{I} \odot (1 - \mathbf{M}_{\text{port}}^{\text{train}}), \tag{11}$$

where $\odot$ denotes element-wise multiplication. Here, $\mathbf{M}_{\text{attr}}$ localizes the original attribute region, while $\mathbf{M}_{\text{port}}^{\text{train}}$ defines the expanded region into which the attribute will be inserted during generation. This expansion process is *attribute-aware* as it preserves the intended attribute category while diversifying its spatial extent. Unlike HairFusion (Chung et al., 2025), which expands masks using fixed heuristics specific to hair, our approach generalizes across multiple facial attributes and enables the model to learn spatially flexible yet semantically grounded transfer patterns.

**Reference image augmentation.** To address the limited diversity of self-reconstruction setups, we introduce an augmentation pipeline that improves robustness to pose, alignment, and appearance variations in attribute–portrait pairs. We perturb both the attribute-only and masked portrait images to simulate realistic spatial and photometric variations. We apply random affine transformations (translation, scaling, rotation) to induce spatial misalignment, and use the FLUX outpainting model (Labs, 2024) to inpaint newly exposed regions. Additionally, color jittering on tone, contrast, saturation, and hue accounts for appearance variations. This strategy exposes the model to diverse configurations, enabling more robust attribute transfer and animation under real-world variations.

### 3.4 INFERENCE FRAMEWORK AND EXTENSIONS

**Inference pipeline.**  At inference time, our system takes as input a portrait image, an attribute image, and a keypoint sequence. We first construct two masked reference images: the attribute-only image $\tilde{\mathbf{I}}_{\text{attr}}$ and the attribute-masked portrait image $\tilde{\mathbf{I}}_{\text{port}}$, by applying segmentation masks predicted by Sapiens (Khirodkar et al., 2024) to the attribute image $\mathbf{I}_{\text{attr}}$ and the portrait image $\mathbf{I}_{\text{port}}$. To improve spatial alignment between the attribute and portrait inputs, we introduce a *Face Aligner* module, which repurposes a lightweight image-to-3D avatar model (Chu & Harada, 2024) solely for alignment. This module reconstructs a coarse 3D avatar from the attribute image and aligns its shape and pose to the portrait using FLAME (Li et al., 2017) parameters $(\boldsymbol{\beta}, \boldsymbol{\theta}, \boldsymbol{\psi})$ estimated by EMOCA (Daněček et al., 2022). From the resulting pose-aligned image $\mathbf{I}_{\text{attr}}^{\text{align}}$, we extract a refined attribute mask $\mathbf{M}_{\text{attr}}^{\text{align}}$ using Sapiens. This mask is then merged with the initial portrait mask $\mathbf{M}_{\text{port}}^{\text{init}}$ to define the final transferable region $\mathbf{M}_{\text{port}}^{\text{infer}} = \mathbf{M}_{\text{port}}^{\text{init}} \cup \mathbf{M}_{\text{attr}}^{\text{align}}$. The updated mask is applied to construct the final attribute-masked portrait image, $\tilde{\mathbf{I}}_{\text{port}} = \mathbf{I}_{\text{port}} \odot (1 - \mathbf{M}_{\text{port}}^{\text{infer}})$, as illustrated in Fig. 3. Finally, spatial features $\mathcal{F}_{\text{attr}}, \mathcal{F}_{\text{port}}$ and semantic embeddings $\phi_{\text{attr}}, \phi_{\text{port}}$ are extracted from the two masked reference images. Conditioned on these features and the keypoint sequence, DNet synthesizes a video of the target identity with the desired attribute through iterative denoising (Eq. (8)).

**Multi-attribute transfer.**  Our model supports zero-shot composition of multiple attributes without additional training, by generalizing the spatial attention formulation in Eq. (5). Instead of using a single attribute feature, we concatenate multiple attribute feature maps along the width dimension:

$$\bar{\mathbf{F}}_t^l = \text{SA}\left(\mathbf{F}_t^l, \mathbf{F}_{\text{port}}^l, \text{concat}_{\text{w}}\left(\mathbf{F}_{\text{attr}}^{l,1}, \mathbf{F}_{\text{attr}}^{l,2}, \cdots, \mathbf{F}_{\text{attr}}^{l,N_{\text{attr}}}\right)\right), \tag{12}$$

where each $\mathbf{F}_{\text{attr}}^{l,k}$ denotes the feature map extracted from the $k$-th attribute-only image using the ARNet. To construct the final attribute-masked portrait in this setting, we also generalize the mask fusion process by taking the union of all aligned attribute masks:

$$\mathbf{M}_{\text{port}}^{\text{infer}} = \mathbf{M}_{\text{port}}^{\text{init}} \cup \bigcup_{k=1}^{N_{\text{attr}}} \mathbf{M}_{\text{attr}}^{\text{align},k}, \tag{13}$$

where each $\mathbf{M}_{\text{attr}}^{\text{align},k}$ is the aligned mask extracted from the $k$-th attribute image. This composite mask is then used to remove all attribute regions from the portrait image before generation. The rest of the attention computation remains unchanged, allowing the model to jointly attend to all attributes and synthesize coherent multi-attribute compositions without retraining.

**Attribute interpolation.**  Our model enables zero-shot interpolation between two attributes of the same category (e.g., hairstyle A and B) without fine-tuning (Zhang et al., 2024a; Cha et al., 2025). Given two attribute-only images, we extract spatially attended features $\bar{\mathbf{F}}_t^{\tau,l,1}$ and $\bar{\mathbf{F}}_t^{\tau,l,2}$ using our spatial attention, and interpolate them as follows:

$$\bar{\mathbf{F}}_t^{\tau,l} = (1 - \alpha)\,\bar{\mathbf{F}}_t^{\tau,l,1} + \alpha\,\bar{\mathbf{F}}_t^{\tau,l,2}, \tag{14}$$

where $\alpha \in [0, 1]$ controls the interpolation ratio. The interpolated feature $\bar{\mathbf{F}}_t^{\tau,l}$ is then passed to DNet for generation. This enables smooth and semantically consistent transitions between attributes.

## 4 EXPERIMENTS

**Experimental setup.**  To address the lack of ground-truth data for cross-identity attribute transfer, we design two evaluation settings: *self-attribute transfer* and *cross-attribute transfer*. In **self-attribute transfer**, a single video is split into a portrait and an attribute image from different frames of the same identity, and the model reconstructs the original video. While useful for controlled evaluation, this provides only a pseudo ground-truth and mainly reflects reconstruction ability rather than the full complexity of cross-identity transfer. In **cross-attribute transfer**, the portrait and attribute

Table 1: **Quantitative Comparison.** We compare our method with recent approaches that (1) synthesize portraits with transferred hairstyles, and (2) animate the synthesized portrait image.

| Img.Gen. | Animation | Self-Attribute Transfer | | | | | Cross-Attribute Transfer | | | | |
|---|---|---|---|---|---|---|---|---|---|---|---|
| | | $L_1 \downarrow$ | PSNR↑ | SSIM↑ | LPIPS↓ | FID↓ | mCLIP-I↑ | mDINO↑ | ID-Sim.↑ | $VFID_{I3D} \downarrow$ | $VFID_{ResNeXt} \downarrow$ |
| PbE | LivePortrait | 0.1059 | 16.14 | 0.5641 | 0.2859 | 40.63 | 0.8499 | 0.6407 | 0.5630 | 37.6462 | 3.3868 |
| | X-Portrait | 0.1180 | 15.33 | 0.5270 | 0.2978 | 59.20 | 0.8393 | 0.5916 | 0.5458 | 36.7030 | 2.9008 |
| | MegActor-$\sum$ | 0.1268 | 14.82 | 0.4840 | 0.3157 | 62.77 | 0.8535 | 0.6266 | 0.4863 | 38.2746 | 6.2743 |
| HairFusion | LivePortrait | 0.1438 | 13.76 | 0.4801 | 0.3792 | 46.24 | 0.8741 | 0.6843 | 0.6502 | 30.5632 | 2.6719 |
| | X-Portrait | 0.1511 | 13.30 | 0.4334 | 0.3733 | 59.02 | 0.8809 | 0.6914 | 0.6520 | 30.2570 | 4.9184 |
| | MegActor-$\sum$ | 0.1650 | 12.75 | 0.4138 | 0.4015 | 65.59 | 0.8736 | 0.6708 | 0.6044 | 30.9702 | 5.3037 |
| StableHair | LivePortrait | 0.1122 | 15.84 | 0.5491 | 0.3041 | 43.74 | 0.8831 | 0.7051 | 0.6564 | 29.5014 | 3.9495 |
| | X-Portrait | 0.1229 | 15.04 | 0.5114 | 0.3117 | 53.36 | 0.8895 | 0.7239 | 0.6443 | 28.2627 | 1.5718 |
| | MegActor-$\sum$ | 0.1301 | 14.62 | 0.4706 | 0.3347 | 63.47 | 0.8848 | 0.7271 | 0.6130 | 30.4087 | **1.4672** |
| TriplaneEdit | LivePortrait | 0.1023 | 16.52 | 0.5511 | 0.2924 | 57.86 | 0.8540 | 0.6163 | 0.2776 | 32.5660 | 8.9103 |
| | X-Portrait | 0.1051 | 16.05 | 0.5401 | 0.2760 | 60.25 | 0.8366 | 0.6216 | 0.2944 | 30.6319 | 2.9315 |
| | MegActor-$\sum$ | 0.1248 | 15.10 | 0.4828 | 0.3293 | 70.41 | 0.8210 | 0.5674 | 0.2770 | 32.5679 | 2.8542 |
| **Ours** | | **0.0744** | **18.83** | **0.6527** | **0.1565** | **38.00** | **0.9043** | **0.7801** | **0.7098** | **27.1547** | 2.4052 |

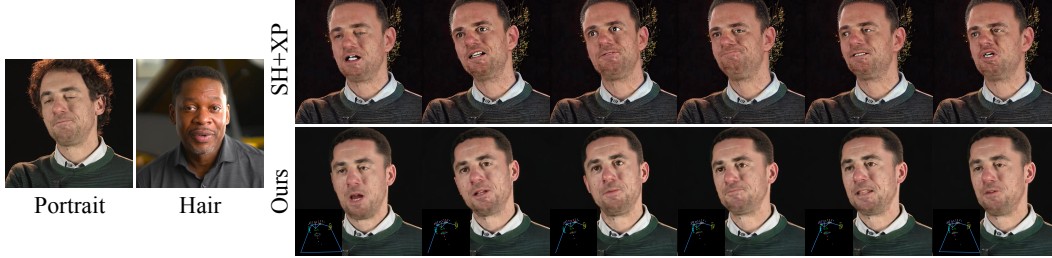

Portrait Animation with Attribute Transfer

Figure 4: **Qualitative Comparison for Cross-Attribute Transfer.** We compare our method and the baselines that combine X-Portrait (Xie et al., 2024) with StableHair (Zhang et al., 2025) in cross-identity transfer setup. We provide more results in our Supp. Mat.

images come from different individuals. Without exact ground-truth, this setting instead evaluates semantic consistency, identity preservation, and temporal realism. Together, the two settings offer a comprehensive evaluation of both low-level fidelity and high-level transfer quality.

**Dataset.** We train our model on CelebV-Text (Yu et al., 2023), VFHQ (Xie et al., 2022), and Nersemble (Kirschstein et al., 2023), totaling 2,747 videos. For evaluation, we sample 200 videos for self-attribute transfer and 50 videos for cross-attribute transfer from CelebV-Text and VFHQ, ensuring diverse and unseen identities, head poses, and expressions. The masks for the portrait and attribute frames are generated following the procedure used in each compared method.

**Metrics.** For self-attribute transfer, we evaluate reconstruction fidelity using $L_1$, PSNR, SSIM, and LPIPS, and perceptual quality with FID (Parmar et al., 2022). For cross-attribute transfer, we measure attribute transfer quality with CLIP-I (Radford et al., 2021; Hessel et al., 2021) and DINO (Caron et al., 2021), identity preservation with ArcFace (Deng et al., 2019), and temporal realism with VFID (Fang et al., 2024) using I3D (Carreira & Zisserman, 2017) and ResNeXt (Hara et al., 2018).

### 4.1 COMPARISON

**Baselines.** As no prior work directly tackles portrait animation with attribute transfer from in-the-wild reference images, we construct two-stage baselines by combining image-level attribute transfer with video animation methods, resulting in 12 model combinations. For attribute transfer (stage 1), we consider: Paint-by-Example (PbE) (Yang et al., 2023), a mask-conditioned diffusion method for reference image insertion; HairFusion (Chung et al., 2025) and StableHair (Zhang et al., 2025), diffusion-based models for hairstyle transfer with and without masks; and TriplaneEdit (Bilecen et al., 2024), a 3D-aware GAN-based face editor. For portrait animation (stage 2), we use: LivePortrait (Guo et al., 2024), X-Portrait (Xie et al., 2024), and MegActor-$\sum$ (Yang et al., 2025).

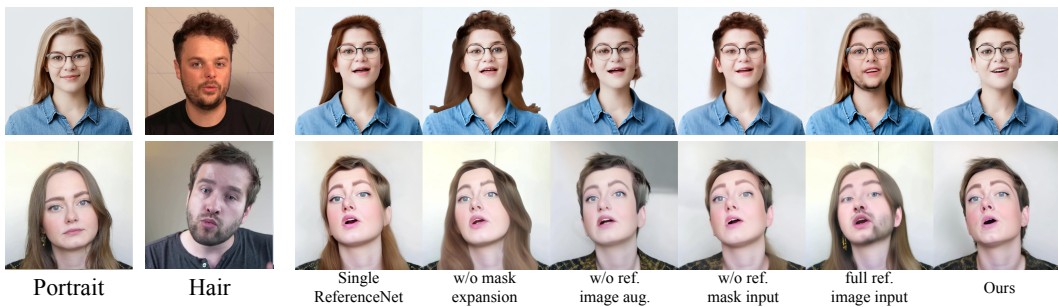

| Portrait | Hair | Single ReferenceNet | w/o mask expansion | w/o ref. image aug. | w/o ref. mask input | full ref. image input | Ours |

Figure 5: **Ablation Study.** Omitting components or altering training scheme degrades visual quality.

**Results.** As shown in Table 1, our method consistently outperforms all baseline combinations across both fidelity and perceptual quality metrics in self-attribute transfer. Fig. 4 presents a qualitative comparison against baselines using LivePortrait (Guo et al., 2024) as the animation module (stage 2). Our method generates coherent and realistic hairstyle animations that preserve the identity and maintain consistency in spatial extent, shape, and fine details across frames. Please refer to our Supp. Mat. for additional qualitative comparisons with other baseline combinations.

## 4.2 ABLATION STUDY

We evaluate the contributions of key components in our model and training strategy. Table 2 presents quantitative results, and Fig. 5 shows corresponding qualitative comparisons. **"single ReferenceNet"** replaces the dual-branch architecture with a shared encoder that receives the portrait and attribute images concatenated along the channel dimension, following CAT-VTON (Chong et al., 2024). This setup fails to separate the roles of the two inputs, resulting in undesired blending of attribute and identity cues. **"w/o mask expansion"** omits the attribute-aware

Table 2: **Ablation Study.** Bold indicates the best, underline the second.

| Variant | $L_1 \downarrow$ | PSNR↑ | SSIM↑ | LPIPS↓ |
|---|---|---|---|---|
| single ReferenceNet | 0.0813 | 17.95 | 0.6314 | 0.1973 |
| w/o mask expansion | 0.0881 | 17.16 | 0.5915 | 0.2073 |
| w/o ref. image aug. | 0.0900 | 16.97 | 0.5973 | 0.2248 |
| w/o ref. mask input | 0.0747 | 18.60 | 0.6511 | 0.1670 |
| full ref. image input | **0.0670** | **19.47** | **0.6698** | **0.1310** |
| **Ours** | 0.0744 | 18.83 | 0.6527 | 0.1565 |

augmentation that simulates variations in spatial extent. Without this strategy, the model tends to rely on the default shape of the portrait's original attribute mask, making it less capable of handling diverse attribute shapes during inference. **"w/o ref. image aug."** disables spatial and photometric augmentations applied to the reference images during training. As a result, the model fails to accurately transfer the desired attribute with misaligned reference images. **"w/o ref. mask input"** removes the binary mask concatenation from the inputs to the ReferenceNets. This weakens spatial localization and often leads to artifacts or residual traces of the original attribute in the output. **"full ref. image input"** uses unmasked portrait and attribute images during training. Interestingly, this variant achieves the best quantitative scores in Table 2, which evaluates the self-attribute transfer setting, since full images simplify the task by allowing the model to copy content more easily. However, as shown in Fig. 5, this model fails to disentangle identity and attribute roles, leading to visible identity leakage during cross-identity transfer. **Ours** achieves spatially consistent, identity-preserving results, and quantitatively outperforms all other ablated variants except the full reference image variant.

## 4.3 APPLICATION

**Multi-attribute transfer.** Our model supports the composition of multiple attributes (*e.g.*, glasses, hat, hairstyle) in a single generation pass by extending the spatial attention mechanism as described in Eq. (12). Fig. 6 shows qualitative results where multiple attributes are simultaneously transferred from different reference images. Remarkably, our model not only combines multiple attributes seamlessly but also handles interactions between overlapping regions, such as between hair and a hat. Despite the reference images exhibiting diverse lighting conditions and spatial alignments, the model

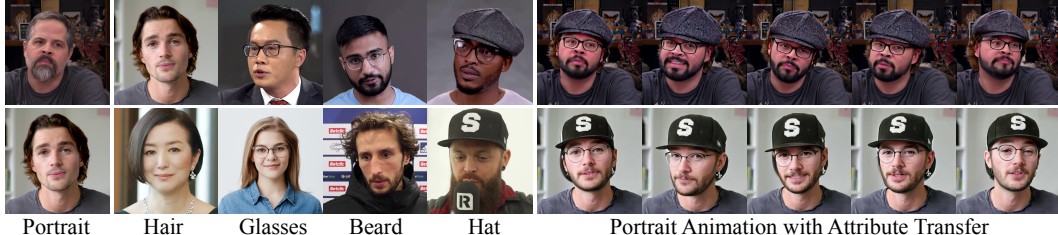

Figure 6: **Multi-Attribute Transfer.** Our model supports composition of multiple attributes (*e.g.*, hair, eyeglasses, beard, hat) in a single forward pass without additional training.

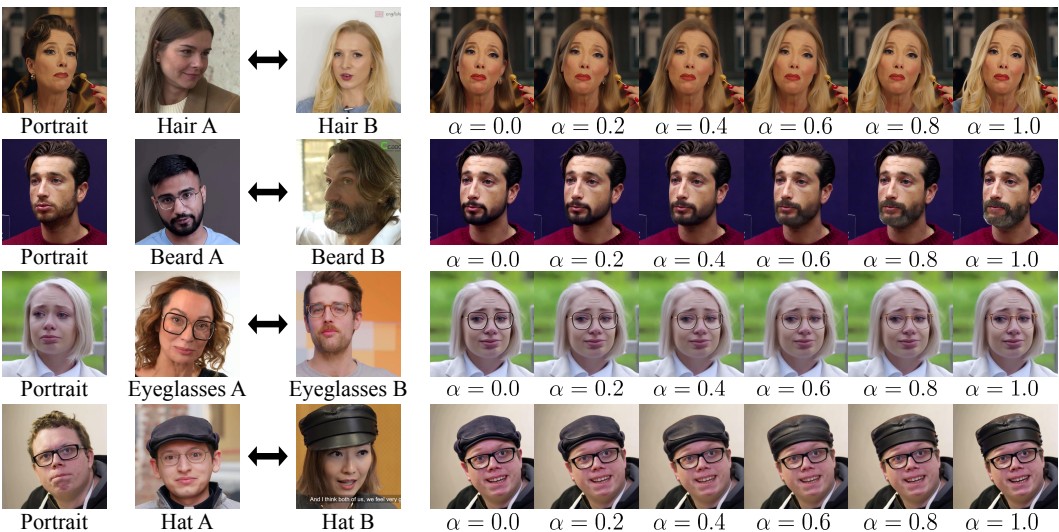

Figure 7: **Attribute Interpolation.** Our method generates zero-shot, single-stage portrait animations with interpolated attributes, even for rigid objects such as hats and eyeglasses. The animations interpolate naturally according to the $\alpha$ values.

successfully integrates all attributes into the portrait image while maintaining a coherent and natural appearance.

**Attribute interpolation.** Our model enables attribute interpolation by linearly blending the reference features of two attributes, as in Eq. (14). Fig. 7 shows the results with smooth transitions in shape and appearance. The interpolations exhibit smooth changes in visual attributes, demonstrating that our model effectively captures semantically meaningful directions in the attribute feature space.

## 5 CONCLUSION

We present Durian, a zero-shot framework for portrait animation with cross-identity attribute transfer, given a portrait image and one or more reference images specifying the target attributes. Our diffusion model, equipped with a Dual ReferenceNet, learns attribute transfer directly from uncurated portrait videos through a self-reconstruction training strategy, eliminating the need for triplet supervision. This is further enhanced by our attribute-aware mask expansion and augmentation scheme. Moreover, Durian naturally extends to multi-attribute composition and attribute interpolation within a single generation pass, without requiring any additional training.

**Acknowledgments.** This work was supported by NRF grant funded by the Korean government (MSIT) (No. RS-2022-NR070498), and IITP grant funded by the Korea government (MSIT) [No. RS-2024-00439854, No. 2022-0-00156, No. RS-2025-25441838, and No. RS-2021-II211343]. H. Joo is the corresponding author.

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

# A    IMPLEMENTATION DETAILS

## A.1    TRAINING DETAILS

We adopt the two-stage training strategy following Zhu et al. (2024). In the first stage, we resize all videos to a uniform resolution of $512 \times 512$ pixels and train with a global batch size of 8 for 60,000 steps. During this phase, all layers except the temporal attention layers are set to be trainable, as the latter are not yet incorporated into the UNet. In the second stage, we insert temporal attention layers into the Denoising UNet (DNet) and train only these newly added layers. This stage uses 24-frame inputs, a global batch size of 8, and also runs for 60,000 steps. For both stages, we fix the learning rate at 1e-5, with each stage requiring approximately three days of training. We train our model using 8 NVIDIA RTX A6000 GPUs. As initialization, we use the UNet checkpoint from Yang et al. (2023), while the temporal attention layers are initialized from Guo et al. (2023). Our training dataset consists of 2,747 samples drawn from CelebV-Text (Yu et al., 2023), VFHQ (Xie et al., 2022), and Nersemble (Kirschstein et al., 2023). Our method focuses on four attribute categories, with the following distribution: **Hair** – 886 samples from CelebV-Text, 935 from Nersemble, and 265 from VFHQ (total 2,086); **Beard** – 253 samples from CelebV-Text; **Eyeglasses** – 279 samples from CelebV-Text; **Hat** – 129 samples from CelebV-Text. On average, each video contains 292 frames.

## A.2    EVALUATION DETAILS

For self-attribute transfer, we randomly sample 200 videos from CelebV-Text (Yu et al., 2023) and VFHQ (Xie et al., 2022), ensuring that these videos contain unseen identities, facial poses, and expressions relative to the training dataset. For cross-attribute transfer, we additionally sample 50 videos. Masks required for image editing baselines are constructed following the procedures provided by the respective authors. To construct cross-attribute transfer pairs, we use the 50 sampled identities and randomly select corresponding face images from VFHQ and CelebV-Text that do not overlap with the training dataset.

We evaluate the results using several metrics. mCLIP-I (masked CLIP-I (Radford et al., 2021; Hessel et al., 2021)) and mDINO (Caron et al., 2021) (masked DINO) assess whether the target attribute is accurately transferred into the generated portrait animation video. To this end, we fill the background of attribute-only images with white and segment the target attribute region from the generated portrait animation video using Sapiens (Khirodkar et al., 2024). We then fill the segmented background with white and compute frame-wise cosine similarity embeddings with CLIP-I and DINO. ID-Sim evaluates identity preservation. Specifically, we mask attribute regions in portrait images by filling them with black, segment the target attribute regions in the generated videos with Sapiens, and replace them with black before computing frame-wise cosine similarity embeddings with ArcFace. Finally, VFID (Video Fréchet Inception Distance) (Heusel et al., 2017; Wang et al., 2018) extends FID to the video domain. Following Fang et al. (2024), we adopt VFID to measure temporal consistency and overall video quality.

## A.3    KEYPOINT GUIDANCE GENERATION

Our model generates portrait animations using a guidance video composed of facial keypoints, as shown in Fig. 2 of our main paper. These keypoints encode entangled facial shape information, such as interocular distance and the relative positions of eyes, nose, and ears. While this rich representation supports accurate animation in self-attribute transfer scenarios, we observe that, in cross-attribute settings, the generated animation tends to follow the facial shape of the guidance video rather than the portrait image. Also, in these real-world scenarios, significant shape and scale discrepancies between the source and the driver can degrade the model's performance. To address this, we propose a method that preserves the portrait's facial shape while transferring only the motion from a different identity. Specifically, we employ LivePortrait (Guo et al., 2024) to generate an animation of the portrait image that maintains its original shape while being driven by the motion in the guidance video. We then extract a facial keypoint guidance video from this animation using Sapiens (Khirodkar et al., 2024), effectively creating a self-reenactment-like scenario that allows our model to operate more reliably. Note that for all quantitative results reported in our paper and tables, we follow the standard self-reenactment setting (Kim et al., 2024a; Morelli et al., 2022). The facial keypoint guidance is extracted directly from the ground-truth videos, not generated by LivePortrait.

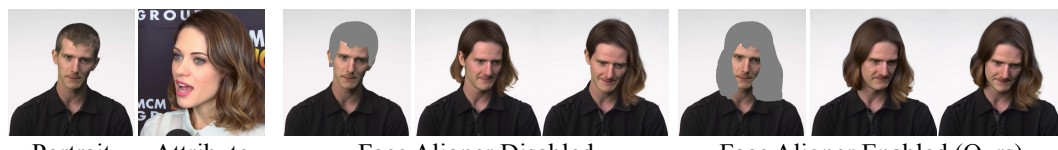

Portrait    Attribute       Face Aligner Disabled       Face Aligner Enabled (Ours)

Figure 8: **Ablation Study for Face Aligner.** Omitting Face Aligner at inference time degrades the visual quality of the generated animation.

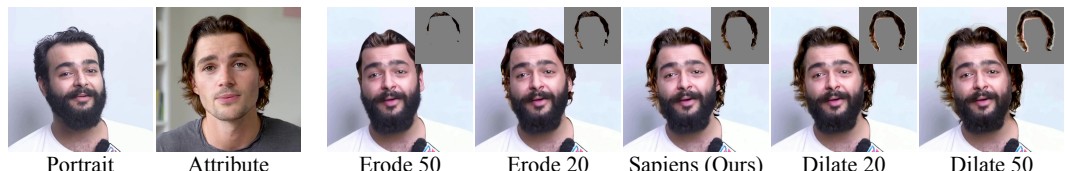

Portrait    Attribute      Erode 50     Erode 20    Sapiens (Ours)    Dilate 20    Dilate 50

Figure 9: **Sensitivity Analysis of the Attribute Mask.** We present an analysis showing how output quality changes with mask quality by applying erosion and dilation to the attribute mask derived from the Sapiens Mask.

## B  ADDITIONAL RESULTS

### B.1  ADDITIONAL ABLATION STUDY FOR FACE ALIGNER

We perform an ablation study on our Face Aligner, as described in Section 3.4 and illustrated in Fig. 3 of the main paper. As shown in Fig. 8, removing Face Aligner still allows the long blonde hair from the attribute image to be transferred to the portrait's target attribute region. However, the generation becomes unstable, with the left hair strand intermittently appearing and disappearing. In contrast, ours, which applies the face aligner at inference time, enables stable transfer, ensuring that the long blond hair remains consistently preserved throughout the animation.

### B.2  SENSITIVITY ANALYSIS OF THE ATTRIBUTE MASK

We show a mask sensitivity analysis by systematically eroding and dilating the hair masks $\mathbf{M}_{\mathrm{attr}}$ in Fig. 9. When the mask is eroded, it no longer fully covers the target hair region, resulting in a spatially shorter transferred hairstyle. Nonetheless, the model still produces a visually plausible hair transfer video. Conversely, moderate mask dilation has little impact on the overall visual quality, indicating robustness to typical boundary uncertainties in real-world segmentation.

### B.3  ADDITIONAL QUALITATIVE COMPARISON

**Qualitative comparison of self-attribute transfer.**  We additionally provide qualitative results with other baseline combinations in a self-attribute transfer setup. Note that we generate portraits with transferred hair attributes using recent image insertion and face editing methods (Chung et al., 2025; Yang et al., 2023; Zhang et al., 2025; Bilecen et al., 2024), and compare the resulting animation videos produced by applying recent animation techniques (Guo et al., 2024; Xie et al., 2024; Yang et al., 2025) with those generated by our method, as shown in Fig. 10.

**Qualitative comparison of cross-attribute transfer.**  We extend the comparison in Fig. 4 of the main paper and present results in Fig. 11 against 12 baselines for cross-attribute transfer setup. Our method best preserves the identity of the portrait image while most accurately transferring the hairstyle from the attribute image. Furthermore, our results are perceived as the most natural and visually coherent.

### B.4  ADDITIONAL QUANTITATIVE COMPARISON

As shown in Fig. 12 and Table 3, we compare TriplaneEdit (Bilecen et al., 2024)+LivePortrait (Guo et al., 2024) with our method, since TriplaneEdit also supports transfer for eyeglasses. Our method

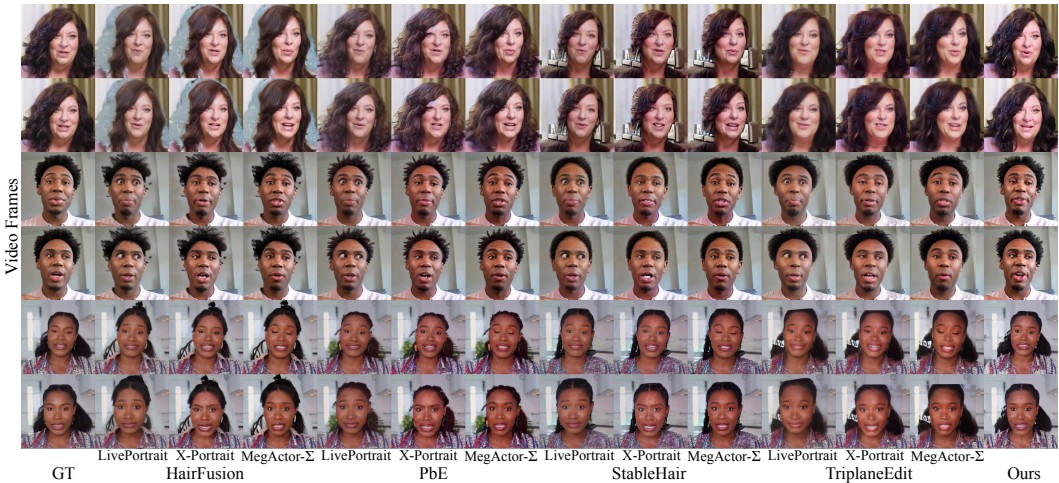

Figure 10: **Qualitative Comparison of Self-Attribute Transfer in the Hair Category.** We compare our method and the baselines that combine portrait animation method with image or hairstyle editing methods. Our results show the highest quality closest to the ground truth, while other methods produce artifacts or unnatural appearances.

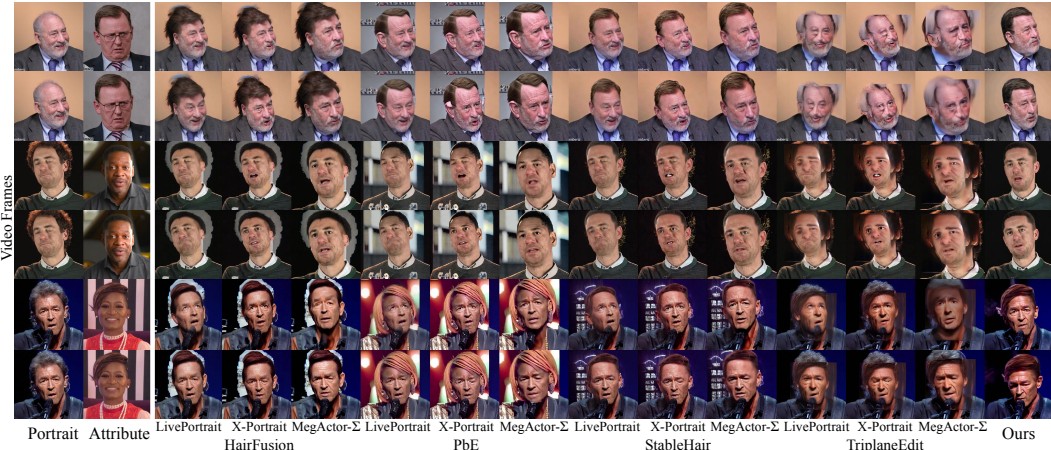

Figure 11: **Qualitative Comparison of Cross-Attribute Transfer in the Hair Category.** We compare our method with the baselines that combine image editing and portrait animation. Our results best preserve the identity of the portrait image while most effectively transferring the hairstyle.

consistently outperforms the baseline across all self-attribute transfer metrics. Moreover, it produces results that are closer to the ground truth and more natural than the baseline.

## B.5 USER STUDY

We conduct a user study to evaluate portrait animations generated using portrait and attribute inputs from different identities, as shown in Table 4. Each of the 100 participants viewed 9 randomly selected videos from a pool of 44 and rated how well each output preserved the hairstyle of the attribute image and the identity of the portrait image. Our method achieves the highest user preference, demonstrating superior performance in cross-identity transfer. Participants were asked: *"Which video most naturally combines the face from the 'face' image with the hairstyle from the 'hair' image?"*

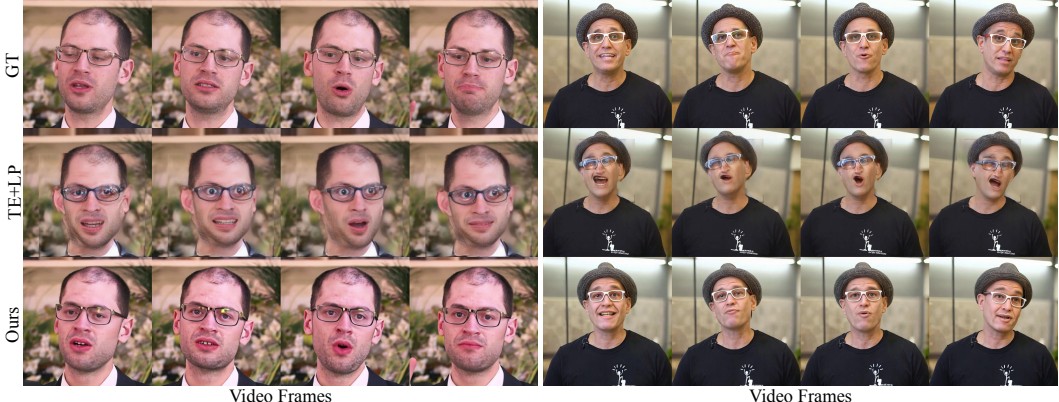

Figure 12: **Qualitative Comparison of Self-Attribute Transfer in the Eyeglasses Category.** TE represents TriplaneEdit and LP denotes LivePortrait. In the self-attribute transfer setting on the eyeglasses category, we compare our results with baseline. Our method produces portrait animations most similar to the ground truth while remaining the most natural.

Table 3: **Quantitative Comparison on Eyeglasses Category.** Our method outperforms this baseline on every evaluation metric.

| Img.Gen. | Animation | $L_1 \downarrow$ | PSNR↑ | SSIM↑ | LPIPS↓ | FID↓ |
|---|---|---|---|---|---|---|
| TriplaneEdit | LivePortrait | 0.151 | 13.53 | 0.433 | 0.435 | 106.28 |
| **Ours** | | **0.078** | **18.19** | **0.627** | **0.181** | **75.59** |

## B.6 ADDITIONAL RESULTS

**Single-attribute transfer.** We extend the results of Fig. 1 in the main paper and present in Fig. 17 animations generated by transferring a single attribute to the portrait. Our method preserves the identity of the portrait image while faithfully transferring the attribute from the attribute image, resulting in natural portrait animations with attribute transfer.

**Multi-attribute transfer.** In Fig. 18 and Fig. 19, we present portrait animations generated by simultaneously transferring two and three attributes in a single stage under the zero-shot setting. Through various combinations of the four supported categories (beard, eyeglasses, hair, hat), our method produces portrait animations where attributes are transferred naturally and with high quality, without any additional optimization.

## B.7 TEXT-TO-IMAGE GENERATED ATTRIBUTE TRANSFER FOR PORTRAIT ANIMATION

Our method generates a portrait animation video with attribute transfer given an image containing the desired attribute. We extend this capability by synthesizing the attribute image directly from a text prompt, enabling text-driven control over the target attribute, as illustrated in Fig. 13. Specifically, we leverage the FLUX (Labs, 2024) text-to-image model to generate realistic attribute images, which are then transferred to the portrait image to produce the final attribute-transferred portrait animation.

## B.8 GENERALIZATION ON CARTOON DOMAIN

We present cartoon-style result in Fig. 14. Despite being trained exclusively on real human video data, our Durian shows strong generalization to the cartoon domain without additional fine-tuning, benefiting from the pretrained diffusion prior.

Table 4: **User Study.** We conduct a user study on two baseline methods that achieve strong performance in both self-attribute transfer and cross-attribute transfer. Our approach receives the highest preference among participants.

| Img.Gen. | Animation | User Study(%)↑ |
|---|---|---|
| TriplaneEdit | LivePortrait | 4.45 |
| PbE | LivePortrait | 19.04 |
| **Ours** | | **76.50** |

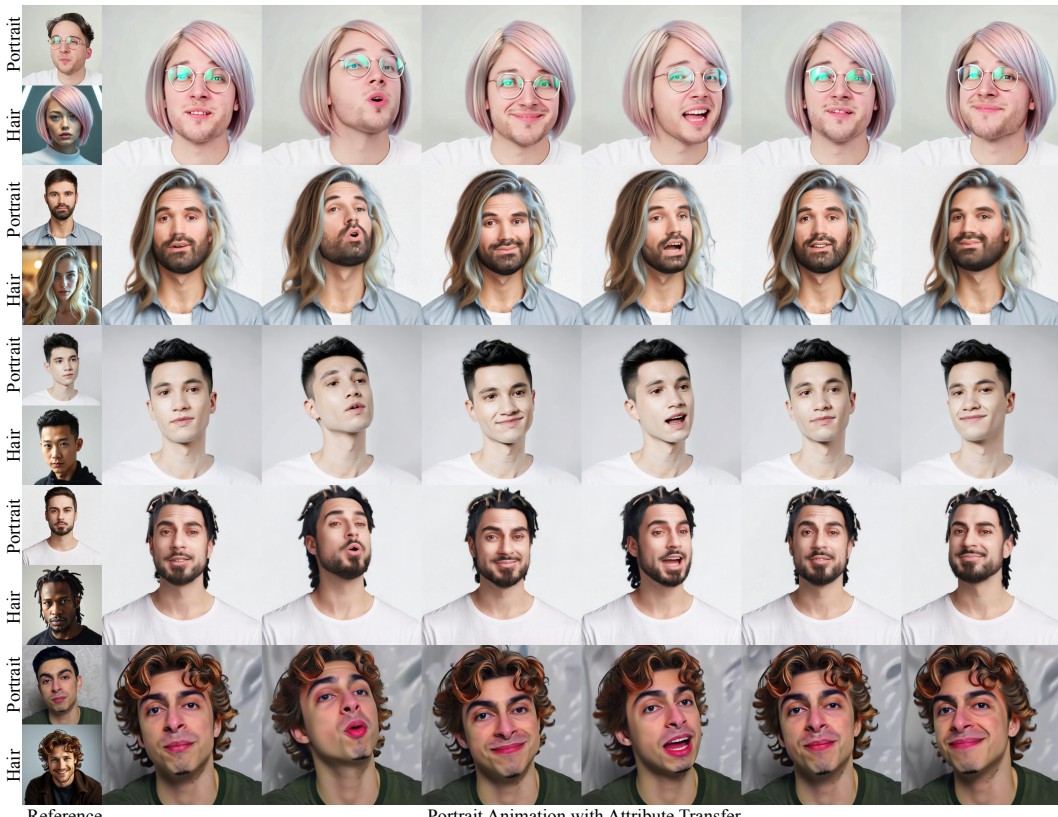

Reference        Portrait Animation with Attribute Transfer

Figure 13: **Text-to-Image Generated Attribute Transfer for Portrait Animation.** We generate a portrait animation with attribute transfer from a textual description by using FLUX (Labs, 2024) to synthesize a high-quality portrait image with the desired hair attribute.

## B.9 FAILURE CASES

**Conflicting Lighting** We present an animation with hair transfer results using a portrait image captured under extremely dark, blue-tinted lighting, while the target hairstyle is taken from a subject photographed outdoors under bright daylight with a white-colored hair appearance. In the resulting animation as shown in Fig. 15, the white hairstyle is transferred accurately; however, the hair appearance does not fully adapt to the portrait's low-light illumination. Nonetheless, we observe that back lighting is partially reflected in the synthesized hair, indicating that the model captures some lighting cues even under severe illumination mismatch.

**Occlusion** We conduct a qualitative experiment on occluded face input as shown in Fig. 16. We demonstrate hair transfer with animation using a portrait image in which part of the face is occluded by a hand with complex manicure patterns. In the resulting animation, minor artifacts appear around the nose region, likely due to the challenging occlusion. Nevertheless, the hair transfer and the generation of the occluded mouth region are successful, and the mouth motion aligns well with the

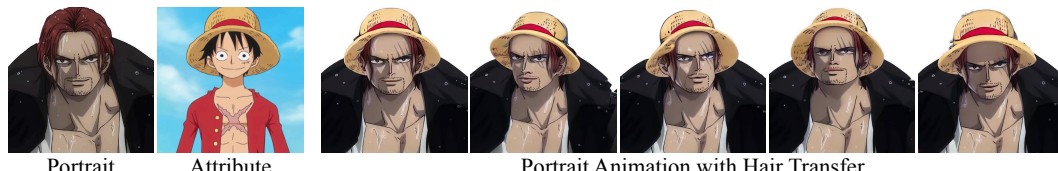

Portrait      Attribute            Portrait Animation with Hair Transfer

Figure 14: **Generalization on Cartoon Domain.** We present our Durian's portrait animation with hat transfer results generated from a cartoon portrait image and a cartoon hat image.

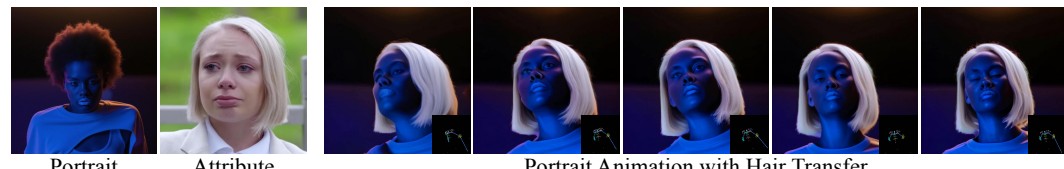

Portrait      Attribute            Portrait Animation with Hair Transfer

Figure 15: **Failure Case under Conflicting Lighting Conditions.** We present a failure case under large lighting discrepancies. Although the white hairstyle is correctly transferred, its color does not adapt to the portrait lighting.

keypoint guidance video, indicating that the model can robustly synthesize motion-consistent facial regions even under partial occlusion.

## C    DISCUSSION

### C.1    EVALUATION IN THE SELF-ATTRIBUTE TRANSFER SCENARIO

In an ideal evaluation, one would use ground-truth videos that contain before and after versions of the attribute transfer. Since such paired data does not exist, the commonly used alternative in attribute transfer and VTON literature is the self-attribute(or garment) transfer setting (Kim et al., 2024a; Chung et al., 2025; Zhang et al., 2025). In this setup, we take a ground-truth video and select two arbitrary frames as the portrait image and the attribute image. The generated video is then compared against the original ground truth video using reconstruction metrics such as L1, PSNR, SSIM, and LPIPS, as reported in Table 1 and Table 2. Although this setting cannot directly evaluate genuine cross-identity transfer, it still provides useful information because the attribute region in the portrait image and the identity region in the attribute image are masked out (see the inputs in Fig. 2). This forces the model to combine complementary cues in order to reconstruct the video.

### C.2    REGARDING FULL REF. IMAGE INPUT VARIANT IN ABLATION STUDY

We provide additional clarification regarding the full reference image input variant presented in the ablation study. The ablation study shown in Table 2 is via self-attribute transfer setting. However, in the "full ref. image input" ablation where masks are not used, the model can simply copy either the portrait or the attribute input because both already contain the required face and attribute cues. As a result, the model can obtain strong reconstruction metrics in Table 2 without learning true disentanglement. Importantly, this shortcut is specific to this variant: for all other ablations, the identity region in the attribute image and the attribute region in the portrait image are masked out, preventing such leakage and ensuring a fair and meaningful comparison. The limitation of the unmasked shortcut becomes evident in cross-attribute transfer, where the model fails to separate identity and attribute cues when the two inputs come from different sources, as shown in Fig. 5. We therefore suggest interpreting Fig. 5 together with Table 2 to understand this contrast.

## USE OF LARGE LANGUAGE MODELS (LLMS)

In accordance with the ICLR policy on the use of Large Language Models (LLMs), we disclose that ChatGPT (OpenAI, 2025) (an LLM developed by OpenAI) was used during the preparation of this

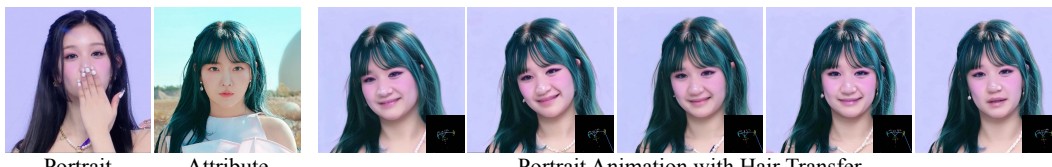

Figure 16: **Failure Case under Portrait Occlusion.** We present a failure case when the portrait image contains occlusions. Artifacts appear around the nose region.

manuscript. The model was employed exclusively for sentence-level grammar checking and minor style corrections.

No parts of the research ideas, methodology, experimental design, or conclusions were generated by the LLM. All scientific contributions are solely attributable to the authors.

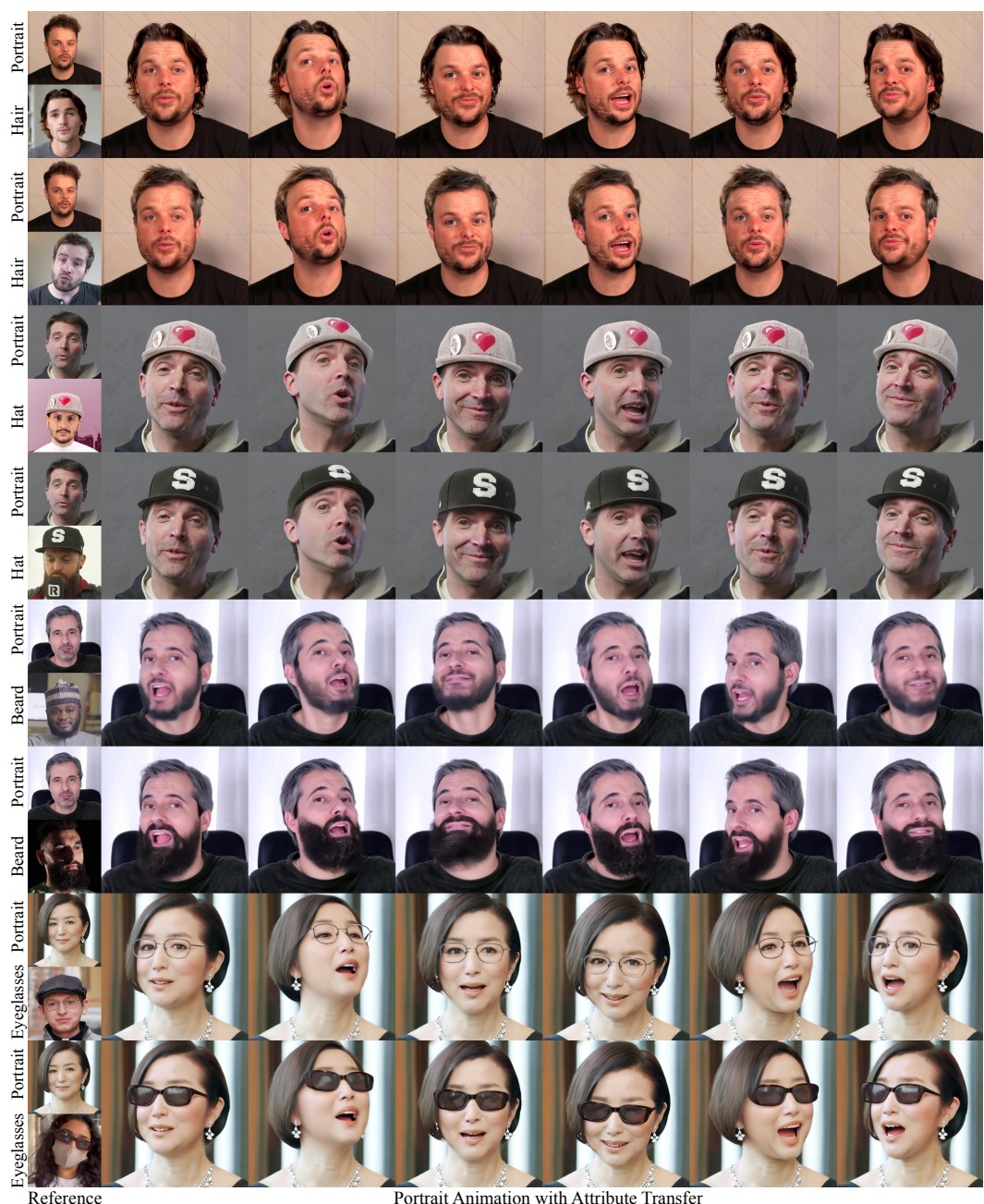

Figure 17: **Qualitative Results for Single-Attribute Transfer.** We present additional results on hair, hat, eyeglasses, and beard attribute transfer for portrait animation. Our method preserves the fine details of the original portrait while achieving natural and seamless attribute transfer.

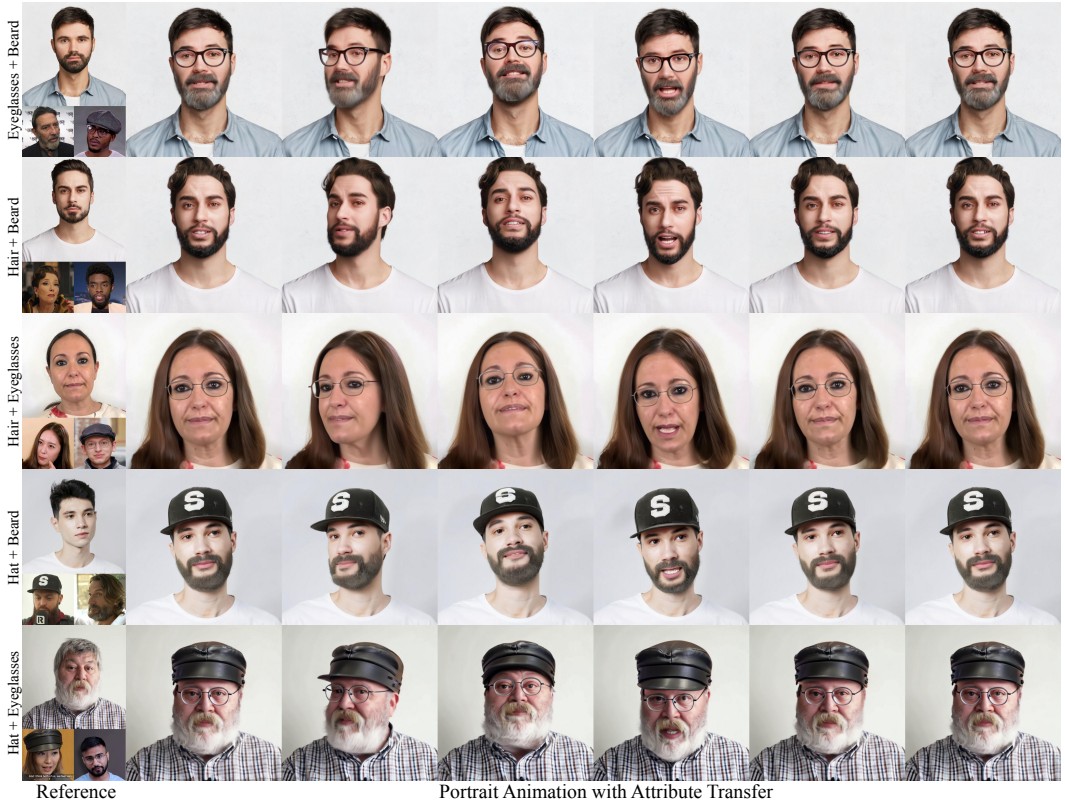

Figure 18: **Qualitative Results for Dual-Attribute Transfer.** We demonstrate the results of simultaneously transferring two attributes for portrait animation.

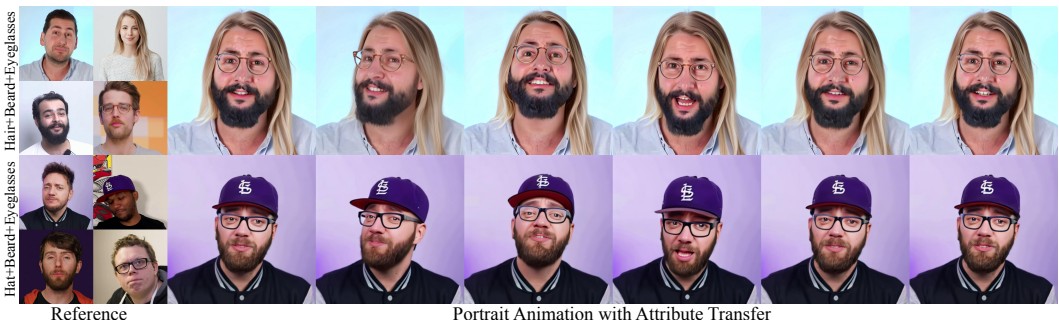

Figure 19: **Qualitative Results for Triple-Attribute Transfer.** We present the results of simultaneously transferring three attributes. In each example, the image in the top-left corner indicates the target portrait.

