# OpenReview forum: "Durian: Dual Reference Image-Guided Portrait Animation with Attribute Transfer"
_ICLR.cc/2026/Conference — ICLR 2026 Poster_

### Official Review · Reviewer_3NqY · 2025-10-24

**Soundness:** 3
**Presentation:** 4
**Contribution:** 4
**Rating:** 8
**Confidence:** 4

**Summary:**

This paper proposes DURIAN, a diffusion-based framework for dual-reference image-guided portrait animation, which enables attribute transfer across identities (e.g., hairstyle, glasses, beard, hat) while maintaining the target person’s identity consistency and generating high-fidelity dynamic videos.The paper introduces four core innovations:
1. Dual ReferenceNet architecture – separately processes the attribute reference image and the identity reference image, and fuses their representations through a Spatial Attention mechanism within the diffusion process.
2. Self-Reconstruction Training Strategy – trains solely on natural portrait videos by sampling two frames from the same video as pseudo attribute–identity pairs, enabling unpaired attribute transfer learning.
3. Attribute-Aware Mask Expansion and Augmentation – bridges the gap between self-reconstruction training and real cross-identity inference, improving robustness to spatial misalignment and diverse attribute layouts.
4. Emergent Abilities – DURIAN exhibits zero-shot multi-attribute composition and interpolation in a single forward pass without any additional training.
Extensive experiments on CelebV-Text, VFHQ, and Nersemble datasets show that DURIAN significantly outperforms multiple attribute transfer baselines both quantitatively and visually.

**Strengths:**

1. Clever self-reconstruction strategy: The proposed self-supervised training method effectively eliminates the dependence on paired attribute data, which is scarce in real-world scenarios, making it an important advancement in attribute transfer.
2. High technical originality: The dual-reference architecture independently encodes identity and attribute information, and fuses them via cross-attention, representing a novel design in diffusion-based frameworks.
3. Robust identity preservation: The introduction of mask expansion and augmentation significantly improves robustness under cross-identity attribute generation, ensuring consistent identity preservation.
4. Excellent writing and structure: The paper clearly motivates the research problem, presents the method with logical flow, and provides solid reasoning for each design choice. The methodology section follows a top-down structure, clearly explaining design rationale and detailed implementations.
5. Detailed equations and reproducibility: The mathematical formulations are clear and complete. The appendix provides comprehensive implementation details, enhancing reproducibility.
6. Comprehensive experiments: Quantitative metrics include both reconstruction fidelity and perceptual quality; Ablation studies verify the effect of each module (mask expansion, augmentation, dual-branch design, etc.); Comparison with strong baselines shows clear superiority.
7. Emergent properties: The model demonstrates emergent multi-attribute composition and smooth interpolation in a single forward pass, indicating strong representational power and generalization.

**Weaknesses:**

1. Handling of overlapping attributes: The paper does not explicitly describe how the model handles the interaction between overlapping attributes such as hair and hat regions. Providing a detailed explanation or visualization of how the spatial attention mechanism resolves these conflicts would make the method more transparent and convincing.
2. Heavy dependency on external tools: The approach heavily relies on multiple external pretrained modules, including Sapiens, EMOCA, FLUX, and SDXL. While this integration improves practicality, it may reduce the model’s reproducibility and independence. Since the mask quality from Sapiens directly affects transfer results (for example, incomplete glasses detection can cause attribute failure), it would be beneficial to conduct a sensitivity analysis or robustness evaluation with alternative segmentation tools.
3. Limited evaluation domain: The current evaluation focuses only on human facial portraits, without testing on stylized or non-human avatars. Including results on cartoon or 3D-rendered portraits would help demonstrate the model’s generalization ability across different visual domains.
4. Empirical design of mask expansion: The mask expansion strategy, which leverages SDXL and ControlNet with prompt engineering to simulate variations such as long hair or reshaped hairstyles, appears somewhat heuristic. It would strengthen the technical rigor to include quantitative results or introduce a more automatic and controllable expansion mechanism.
5. Lack of semantic consistency validation in interpolation: The interpolation experiments (Fig. 7) only show visual examples, without verifying whether the interpolation path corresponds to a semantically smooth transformation. Incorporating semantic distance or latent-space curvature analysis would help confirm that the interpolation indeed reflects continuous attribute transitions.

**Questions:**

1. Could the proposed method be extended to audio-driven scenarios, where an attribute reference is applied to a portrait and animated according to an input audio signal?
2. How does the model handle overlapping attributes during multi-attribute transfer (e.g., hat and hair regions)?
3. Can the model generalize to non-human domains such as stylized or 3D avatars?
4. How sensitive is DURIAN to the accuracy of Sapiens masks? Does noisy segmentation significantly degrade results?
5. Could the self-reconstruction setup be extended to multi-frame attribute co-training (e.g., three or more frames to jointly learn multiple attributes)?
6. Could the authors provide examples of failure cases under extreme conditions (occlusion, conflicting lighting, overlapping attributes, etc.)?

---

> ### Author Response · Authors · 2025-11-25
> **Response to Reviewer 3NqY (1/2)**
>
> **[W4] Design of mask expansion**
>
> We thank the reviewer for the constructive comment.
> We would like to clarify that our mask expansion strategy is not a hand-crafted, sample-specific heuristic, but an automatic and stochastic data augmentation pipeline.
>
> First, its effectiveness is quantitatively validated in Tab.2 under the ablation setting "w/o mask expansion" and qualitatively illustrated in Fig.5.
> Without mask expansion, the model tends to overfit to the default shape of the original attribute mask, which limits its ability to generalize to diverse attribute geometries at inference.
> By exposing the model to a broad distribution of synthetic shape variations during training, mask expansion mitigates such shape bias in our self-reconstruction setup.
>
> Second, the pipeline is fully automatic and does not rely on manually designed prompts for individual cases.
> We construct a dictionary of descriptive attribute modifiers (e.g., long, short, wavy, curly) and randomly sample combinations to generate prompts for SDXL + ControlNet.
> This stochastic prompt generation enables systematic and scalable mask variation without any human intervention.
>
> Finally, our approach is not limited to the hair category.
> Unlike HairFusion [Chung et al., 2025], which uses hair-specific heuristics, our pipeline generalizes to multiple facial attributes, enabling the model to learn spatially flexible yet semantically grounded transfer patterns across attributes, as also discussed in L294--296 of the main paper.
> We will further expand this discussion with a more detailed description of the automatic mask expansion pipeline and its design rationale in the revised manuscript.
>
>
> **[Q1] Extension to audio-driven scenarios**
>
> Thank you for your insightful suggestion. In our current framework, facial keypoint images are encoded by the pose encoder and then passed to DNet to drive the animation. Replacing these keypoints with features extracted from an audio-driven encoder would make audio-conditioned animation feasible within the same architecture. This is an exciting direction for future work.
>
>
> **[Q2 and W1] Handling overlapping attributes**
>
> Thank you for your thoughtful question. As noted in Supplemental Material Line 768, our model is finetuned from a diffusion prior that has already learned strong spatial correlations between hair and hat regions—specifically, that hats consistently appear on top of hair in the training distribution. Because Durian performs attribute transfer while generating portrait animation, it inherits this strong prior during finetuning. This allows the model to handle overlapping attributes in a stable manner, following the learned spatial ordering of these regions.
>
>
> **[Q3 and W3] Generalize to non-human domains**
>
> Thank you for your insightful suggestion. We include cartoon-style result in the 3224\_rebuttal.mp4 from 00:16 to 00:25.
> Despite being trained exclusively on real human video data, Durian shows strong generalization to the cartoon domain without additional fine-tuning, benefiting from the pretrained diffusion prior. We will include these results in the revised manuscript.

---

> ### Author Response · Authors · 2025-11-25
> **Response to Reviewer 3NqY (2/2)**
>
> **[Q4 and W2] Mask sensitivity**
>
> We compare the output quality under different mask qualities in 3224\_rebuttal.mp4 from 00:26 to 00:36. While higher quality masks generally produce the most natural results, the outputs from the Erode 20 and Dilate 20 settings demonstrate that the model still generates natural portrait animation with attribute transfer even when the masks contain moderate noise. When the mask is eroded more, it no longer fully covers the target hair region, resulting in a spatially shorter transferred hairstyle. Nonetheless, the model still produces a visually plausible portrait animation with attribute transfer.
> Conversely, moderate mask dilation has little impact on the overall visual quality, indicating robustness to typical boundary uncertainties in real-world segmentation scenarios.
>
> **[Q5] Multi-attribute co-training**
>
> Thank you for the thoughtful suggestion. Our self-reconstruction framework and model architecture are, in principle, capable of learning multiple attributes jointly. However, masking many attributes simultaneously at the early stages of training causes substantial loss of identity information, which makes optimization unstable. In addition, the number of masked attributes varies across batches, creating difficulties for scalable training. That said, we agree that the multi attribute joint training strategy proposed by the reviewer is a promising direction, and we believe it has strong potential to further improve the results.
>
> **[Q6] Failure cases under extreme conditions**
>
> We present the failure cases of Durian in 3224\_rebuttal.mp4 from 00:47 to 01:06. When occlusions are present in the portrait, artifacts from the hand region, such as patterns near the fingernails, remain on the nose in the portrait animation results. Furthermore, when the lighting conditions of the portrait image and the hair image are extremely different, the hair color is largely preserved and does not adapt to the target portrait lighting, revealing a limitation in handling severe lighting mismatches. We will explicitly incorporate these observations into the revised manuscript as directions for future work.

---

### Official Review · Reviewer_VRMj · 2025-10-25

**Soundness:** 3
**Presentation:** 2
**Contribution:** 2
**Rating:** 4
**Confidence:** 4

**Summary:**

This paper proposes Durian, a diffusion-based framework for generating portrait animation videos with cross-identity facial attribute transfer (e.g., hairstyles, eyeglasses, beards, hats) from one or more reference images. A core challenge in this field is the scarcity of large-scale paired data (same individual with/without target attributes), which limits existing methods to specialized tasks (e.g., hairstyle-only transfer) or multi-stage pipelines. Durian addresses this by introducing a self-reconstruction training strategy that leverages uncurated in-the-wild portrait videos (CelebV-Text, VFHQ, Nersemble) instead of paired data.

**Strengths:**

- Existing attribute transfer methods (e.g., HairFusion, Chung et al. 2025; StableHair, Zhang et al. 2025) rely on attribute-paired data or synthetic pipelines (e.g., generating bald portraits to pair with hairstyles). In contrast, Durian’s self-reconstruction strategy uses uncurated videos (2,747 total) to simulate attribute transfer, eliminating the need for expensive paired data. This makes it scalable to diverse attributes without dataset curation.
- Single-reference models (e.g., CAT-VTON, Chong et al. 2024; TriplaneEdit, Bilecen et al. 2024) concatenate attribute and identity inputs into a shared encoder, leading to attribute-identity blending. Durian’s ARNet (attribute) and PRNet (identity) separate feature extraction, enforced by complementary masking. Ablation studies confirm this design outperforms single ReferenceNet variants (L₁: 0.0744 vs. 0.0813; SSIM: 0.6527 vs. 0.6314).
- Most SOTA methods are task-siloed: static attribute transfer (e.g., Stable-Makeup, Zhang et al. 2024b) cannot animate, while animation models (e.g., LivePortrait, Guo et al. 2024; X-Portrait, Xie et al. 2024) cannot transfer attributes. Durian uniquely enables multi-attribute composition (e.g., combining hair + glasses + hat) and attribute interpolation (smooth hairstyle transitions) in one generation pass—critical for real-world styling applications (e.g., virtual try-ons).
- The self-reconstruction paradigm inherently suffers from a domain gap (training on single identities vs. inferring across identities). Durian’s mask expansion (union of original/generated attribute masks) and augmentations (affine misalignment, FLUX outpainting) outperform fixed-heuristic methods (e.g., HairFusion’s hair-specific mask expansion). This explains its strong cross-identity results (ID-Sim: 0.7098 vs. baseline 0.2776).

**Weaknesses:**

- Durian only supports four attributes (hair, eyeglasses, beard, hat), narrowing its applicability compared to methods that handle makeup (Stable-Makeup, Zhang et al. 2024b), clothing (AnyFit, Li et al. 2024), or accessories beyond hats (e.g., necklaces). The paper provides no rationale for excluding these attributes, nor discusses modifications needed for deformable attributes (e.g., scarves) or subtle attributes (e.g., lipstick).
- Durian relies heavily on third-party tools for critical components:
Mask generation: Sapiens (Khirodkar et al. 2024) for attribute segmentation. Outpainting: FLUX (Labs, 2024) to fill regions after affine transformations. Keypoint alignment: LivePortrait (Guo et al. 2024) to fix shape mismatch in cross-identity inference. This lack of end-to-end integration limits deployment flexibility and introduces potential errors (e.g., Sapiens failing to segment small attributes like thin eyeglasses).
- The two-stage training (60,000 steps per stage, ~6 days on 8 NVIDIA RTX A6000 GPUs) is computationally expensive. For comparison, single-stage animation models (e.g., Animate Anyone, Hu 2024) train faster while maintaining temporal consistency. The paper does not explore optimization (e.g., adaptive step scheduling) to reduce training time, which is a barrier for real-world adoption.
- Insufficient Theoretical and Ablation Depth
  - The “full ref. image input” ablation (unmasked training) achieves better self-transfer metrics (L₁: 0.0670 vs. Durian’s 0.0744) but fails in cross-identity transfer. However, the paper does not analyze why unmasked inputs break disentanglement (e.g., do they allow the model to copy identity cues instead of learning transfer?).
  - Ablations for temporal attention (critical for animation) are missing. The paper only confirms temporal layers improve consistency but does not quantify their impact (e.g., VFID without temporal layers).
- Durian is only evaluated on 3-second clips. For longer videos (e.g., 10+ seconds), its temporal attention may struggle with drifting attributes (e.g., hairstyle shape changing over time). SOTA video diffusion models (e.g., VideoCrafter2, Chen et al. 2024a) use factorized spatiotemporal attention to maintain long-horizon consistency, which Durian does not adopt.
- For blurred images or unrecognizable faces (e.g., low resolution, occlusions), Durian fails to extract valid attribute masks or align references. This is a critical gap compared to robust methods like HunyuanPortrait (Xu et al. 2025), which uses an enhanced appearance encoder to handle low-quality inputs.
- When combining attributes from references with large pose differences (e.g., a hat from a left-angled face and glasses from a front-facing face), Durian suffers from misalignment. Unlike MimicMotion (Zhang et al. 2024b), which uses pose-aware guidance to align multi-reference features, Durian lacks explicit pose correction for compositional tasks.

**Questions:**

- Can Durian be extended to deformable attributes (e.g., clothing, scarves) or subtle attributes (e.g., makeup)? What modifications to masking (e.g., dynamic masks for moving clothing) or feature extraction (e.g., finer-grained ARNet layers) would be required?
- If external tools (e.g., Sapiens, LivePortrait) are replaced with a built-in masker/aligner, how would performance change? Is end-to-end integration feasible without sacrificing quality?
- Cross-identity transfers with large lighting differences (e.g., a dark-haired reference and a bright-portrait identity) produce subtle artifacts. Would adding an illumination alignment module (e.g., as in PERSE, Cha et al. 2025) improve attribute-identity coherence?
- Training data is imbalanced (2,086 hair samples vs. 129 hat samples). Does this imbalance hurt transfer quality for rare attributes (e.g., hats)? Could few-shot learning or data augmentation for rare attributes improve generalization?

---

> ### Author Response · Authors · 2025-11-25
> **Response to Reviewer VRMj (1/3)**
>
> **[W1] Computationally expensive**
>
> We acknowledge the reviewer’s concern on training cost. However, we would like to emphasize that Durian tackles a strictly more challenging problem than pure video animation, namely, simultaneous facial attribute transfer and temporally coherent animation under a unified framework.
>
> While single-stage animation models such as Champ [1] focus solely on animation, our model must additionally learn fine-grained disentanglement and recomposition of identity and attribute cues, which introduces extra training complexity.
> Notably, Champ adopts a similar two-stage training schedule with $60{,}000 + 20{,}000$ steps.
> Given that our method addresses a strictly more complex objective while using a comparable training budget, we believe the computational cost of Durian is reasonable in the context of recent diffusion-based video models.
>
> While our current focus is on demonstrating the feasibility of the proposed framework, we agree that improving training efficiency is an important direction for future work.
>
> [1] Zhu, Shenhao, et al. "Champ: Controllable and consistent human image animation with 3d parametric guidance." European Conference on Computer Vision. Cham: Springer Nature Switzerland, 2024.
>
>
>
> **[W2] Insufficient theoretical and ablation depth**
>
> Regarding the "full reference image input" baseline, we do provide an analysis of this behavior in Line 457, where we explain why unmasked inputs lead to failure cases such as identity leakage, as shown in Fig. 5. In short, unmasked training allows the network to overfit to appearance cues tied to the reference identity, preventing proper disentanglement between identity and transferred facial attributes. This results in stronger self-transfer metrics but significantly degraded cross-identity transfer performance. We will clarify this explanation further in our revision.
>
> We also thank the reviewer for suggesting additional ablations on temporal attention. In our work, we follow prior ReferenceNet-based architectures for temporal modeling, and these temporal layers are not the core part of our contribution. Our main focus is on attribute transfer, and across the variants of our method in the ablation studies, the degree of temporal coherence remains similar, as shown in the comparison videos provided in our supplementary material.
>
>
> **[W3] Evaluation using 3-second clips**
>
> We respectfully note that the reviewer's concern about "evaluating on 3 second clips only" is incorrect, and no such setup appears anywhere in our paper or supplemental material. We never state or imply that Durian is evaluated on 3 second videos. There is no such assumption in our method or evaluation protocol.
>
> The videos used in our experiments, sampled from CelebV Text and VFHQ, are typically around 10 seconds in duration, and all evaluations are conducted on these full length sequences.
>
> To clarify further, our method does not impose any restriction on video length. In the supplemental video, we present results across a range of durations. For instance, single attribute transfers such as hair or hat modifications run for roughly 5 seconds, multi attribute transfers for about 7 seconds and attribute interpolation sequences for approximately 10 seconds.

---

> ### Author Response · Authors · 2025-11-25
> **Response to Reviewer VRMj (2/3)**
>
> **[W4] Testing on blurred images or unrecognizable faces**
>
> We would like to clarify that our method does not extract attribute masks as outputs. Instead, we rely on an off-the-shelf model (Sapiens) to generate masked attribute images, which are provided as inputs to our system. Our setting focuses on controlled attribute transfer, which has different goals than the broader portrait animation scenario tackled by methods such as HunyuanPortrait [Xu et al. 2025].
>
> Although handling severely degraded inputs lies outside the primary scope of controlled attribute transfer (since such inputs are uncommon in practical usage), we agree that this is a valuable line of analysis. Following the reviewer’s feedback, we have conducted additional qualitative experiment on occluded face input (3224\_rebuttal.mp4 from 00:46 to 00:56). These new results, along with a focused discussion of our method’s current limitations and potential future extensions, will be included in the revision.
>
>
> **[W5] Testing on large pose differences**
>
> We respectfully note that the reviewer’s concern regarding "Durian suffering from misalignment with large pose differences" does not reflect our actual results. The behavior described in the reviewer's comment is not observed in our experiments.
>
> As shown in the supplemental video, Durian demonstrates strong cross-pose alignment even when attributes are taken from faces with substantial pose variation. For example, in the beard results (around 8 seconds in supplemental video), the beard extracted from a clearly right-tilted reference is animated consistently across the full sequence. Likewise, in the hair results (around 36 seconds), a hairstyle taken from a subject facing to the right is transferred and animated coherently over a wide range of target poses. These examples demonstrate that Durian effectively handles pose differences in multi-reference attribute transfer and does not exhibit the misalignment issues suggested in the comment.

---

> ### Author Response · Authors · 2025-11-25
> **Response to Reviewer VRMj (3/3)**
>
> **[Q1] Can Durian be extended to deformable attributes or subtle attributes?**
>
> Our experiments focus on the attribute categories supported by the Sapiens model, including hair, hat, beard, and eyeglasses. However, as the reviewer suggests, attributes such as clothing or scarves can be segmented using existing models like SegFormer or SAM2, making extension to these categories feasible. This flexibility is possible because Durian is designed to be category-agnostic. Whether the attribute involves static or moving clothing is immaterial: Durian requires only a single attribute mask image as input. No architectural modification or dynamic masking mechanism is needed for such extensions.
>
>
> **[Q2] external tools are replaced with a built-in masker/aligner**
>
> The off-the-shelf tools mentioned serve only to prepare inputs for Durian, and the current performance is already achievable using publicly available open-source models. As these external models continue to improve, we expect Durian’s overall performance to benefit correspondingly. We do not propose or rely on any built-in masker or aligner within Durian, and replacing these tools with an integrated module is outside the scope of our work.
>
>
> **[Q3] Cross-identity transfers with large lighting differences**
>
> As demonstrated throughout the supplemental video, Durian already handles significant lighting differences between reference images and portrait identities while producing natural portrait animation with consistent attribute transfer. The types of lighting mismatches mentioned by the reviewer are fully within the range our model manages without requiring an explicit illumination alignment module.
>
> We also note that the cited work by reviewer, PERSE [Cha et al.], does not incorporate any illumination alignment component in its design, so it does not directly address the concern raised in the question.
>
>
> **[Q4] Training data is imbalanced**
>
> Our results across the figures and supplemental videos show that the current dataset distribution is sufficient for Durian to generalize well, even for rarer attributes such as hats. This is possible for two reasons. First, as noted in Supplemental Material Line 768, the UNet prior already contains correlations between the portrait region and hair/hat structures, providing a strong foundation for learning these attributes even with fewer samples. Second, hair and hats occupy highly similar spatial regions on the face, enabling Durian to effectively leverage shared spatial priors when learning transfer and animation behaviors. As a result, we do not observe degradation in hat-transfer quality despite the dataset imbalance.

---

### Official Review · Reviewer_2dwE · 2025-11-01

**Soundness:** 4
**Presentation:** 4
**Contribution:** 4
**Rating:** 8
**Confidence:** 4

**Summary:**

This paper introduces portrait animation system while transferring facial attributes based on reference images. To avoid the attribute-paired dataset issues, it propose self-reconstruction strategies by designing dual reference branches upon diffusion models, significantly outperforming existing SoTAs.

**Strengths:**

- High fidelity on attribute transfer: The proposed system exhibits significantly faithful outcomes on attribute transfer with high quality and relatively no visual artifacts.
- Single generation pass for multiple attributes: For multiple attributes transfer, the concat-based module produces pretty plausible results.

**Weaknesses:**

- Discussion about masking step: It is well known that extracting particular pixels alongside given attributes often fails within in-the-wild condition. It is better to discuss how such scenarios can be further alleviated for practicality.
- Width-wise concatenations: The system propose width-wise concatenation to mix the reference information from both paths in main denoising process. However, in reference-based editing manner, there are some common approaches such as cross attention, per-layer attention map swapping and so forth. It is recommended to discuss how the proposed width-concatenation works compared to existing approaches.

**Questions:**

In multi-attributes transfer, the main component is to concatenate the feature $F^{l, N_{attr}}_{attr}$ with attention module. However, in general, projecting features into key and value vectors relies on learnable projection layer. Since multiple-attributes transfer increase the width of the features thus requiring variable size of linear projections, how the model handles this?

Do the system use separate $N_{attr}$-specific models than single attribute transfer or use some inference-level adaptive manner in single generation pass?

Also, it is recommended to discuss is there any degradation according to the $N_{attr}$ and the maximal number for this.

---

> ### Author Response · Authors · 2025-11-25
> **Response to Reviewer 2dwE (1/2)**
>
> **[W1] Mask under in the wild conditions**
>
> Thank you for raising this point. The main failure mode of the masking step occurs when fine hair strands are not fully segmented. To mitigate this, we combine the Sapiens mask with complementary segmentation models such as RVM [Lin et al.], which provide strand-level hair matting and help recover missing thin structures. We believe that integrating such multi-source masks is a practical and effective way to improve robustness under in-the-wild conditions.
>
> In addition, we conducted a mask sensitivity analysis (see the 3224_rebuttal.mp4 00:26) by systematically eroding and dilating the hair masks. When the mask is eroded, it no longer fully covers the target hair region, resulting in a spatially shorter transferred hairstyle. Nonetheless, the model still produces a visually plausible hair transfer video. Conversely, moderate mask dilation has little impact on the overall visual quality, indicating robustness to typical boundary uncertainties in real-world segmentation.
>
>
> **[W2] Width-wise Concatenations**
>
> We thank the reviewer for the insightful comment.
> Cross-attention–based approaches (e.g., IP-Adapter–style methods) inject extracted features from a reference image, which often lose fine-grained details that are crucial for attribute transfer and video animation. The effectiveness of ReferenceNet-based feature propagation for preserving identity details has already been demonstrated in Animate Anyone [Hu et al., 2024], and we extend this paradigm to a Dual ReferenceNet formulation to simultaneously preserve fine-grained identity and attribute information while enabling their effective disentanglement.
>
> Per-layer feature swapping can be effective for pure animation tasks with a single reference. However, in our attribute transfer setting with two reference sources (portrait and attribute), it is non-trivial to decide how and where to swap features from multiple inputs.
>
> In contrast, our spatial attention operates on width-wise concatenated feature maps that form a unified spatial context across the current frame, portrait, and attribute references. This allows the model to adaptively select and fuse features from both sources, while naturally supporting multi-attribute composition by simply appending additional reference features.

---

> ### Author Response · Authors · 2025-11-25
> **Response to Reviewer 2dwE (2/2)**
>
> **[Q1] Projection layer in multi-attribute transfer**
>
> We appreciate the reviewer’s concern regarding dimensionality growth in multi-attribute transfer.
> As described in Line 222 and Eq.(4), feature concatenation increases the spatial width to
> $c_l \times h_l \times (2 + n) w_l$, where $n$ is the number of attributes, while the channel dimension $c_l$ remains unchanged. Since self-attention performs channel-wise projections (via $W_Q$, $W_K$, and $W_V$) and applies attention over the spatial dimension, increasing $n$ only enlarges the number of spatial tokens without requiring any architectural modification. Since the query is always the denoised video latent, the output of our spatial attention has exactly the same dimensions as the video latents, regardless of the number of concatenated reference features.
> Thus, multi-attribute scaling does not change the representation size or interface of the denoising stream.
>
>
> **[Q2] Does the system require separate models?**
>
> The system operates entirely with a single set of model weights. Our Durian supports single-attribute transfer, multi-attribute transfer, and attribute interpolation without requiring separate models for different numbers of attributes. All results and evaluations presented in the main paper, supplemental material, and supplemental video are generated using this single unified model at inference.
>
>
> **[Q3] Is there performance degradation as the number of attributes increases?**
>
> Thank you for raising this point.
> Our method supports multi-attribute transfer for up to four attribute categories (hair, beard, hat, and eyeglasses) defined in our framework.
> We provide additional results in the supplementary video showing attribute composition with an increasing number of attributes on the same portrait.
>
> Empirically, we do not observe significant degradation in visual quality as more attributes are introduced.
> However, when multiple large attribute regions are combined, a mild identity shift can occur due to the expanded masked area, which limits the available identity context.
> Overall, the model remains stable and produces visually coherent results within the scope of our defined attribute set.

---

### Official Review · Reviewer_5oxU · 2025-11-01

**Soundness:** 2
**Presentation:** 2
**Contribution:** 2
**Rating:** 2
**Confidence:** 5

**Summary:**

This paper presents Durian, which generates portrait animation videos with cross-identity attribute transfer from one or more reference images to a target portrait

**Strengths:**

The task may be new and the method does not need paired data.

**Weaknesses:**

- Figure 2 shows the proposed method use the guidance video frames only. But in A.3, it is stated that the methods use LivePortrait to generate a "self-reenactment" video. This video preserves the shape of the target portrait while matching the movements of the driving video.
Then, extract key points from this intermediate generated video, which will serve as the actual input for the Durian model. The proposed Durian seems to rely on the ability of LivePortrait on the self-reenactment video. The comparison with LivePortrait may not be fair.

- “full ref. image input” uses unmasked portrait and attribute images during training. This variant achieves the best quantitative scores in Table 2. The paper provide explanation in Fig 5. This may show the used metrics are not reliable and the performance in Table 1 is in question. In Figure 10 shows liveportrait with StableHair performs better.

- The paper mention that "no previous work has directly addressed this task". Therefore, it constructs a "two-stage" baseline: Using an image editing model (such as StableHair) to transfer attributes, and then using a video animation model (such as LivePortrait) to create animations. Such comparison is unfair. The two-stage method inherently has disadvantages (error accumulation, domain mismatch). It is expected that the authors' end-to-end model outperforms these "patched-together" baselines in terms of metrics. This is not a solid contribution.

- The core critique of the novelty of the Durian architecture lies in the fact that its "Dual ReferenceNet" is a combination of Adapters. The core mechanism of IP-Adapter (Ye et al., 2023) is as follows: it uses a (typically frozen) image encoder (CLIP) to extract features from a reference image, and injects these features into the cross-attention layers of a pre-trained diffusion model. This enables additional control over the styel.Durian’s ARNet performs exactly the same function. It takes an attribute reference image (hairstyle), encodes it also with CLIP , and injects its spatial featureinto the diffusion U-Net to control the style again. Besides, InstantID (Wang et al., 2024) designs IdentityNet that integrates facial and key-point images into the generation process by imposing strong semantic and weak spatial conditions, thereby achieving high-fidelity id preserve. Durian also performs an identical function. It takes a portrait reference image, encodes it also with ArcFace and injects its feature Diffusion to ensure identity preservation. It seems such two branches are, in fact, a combination of existing adapters.


## Comprehensive Analysis
The so-called "novel architecture" of Durian is essentially a **parallel combination** of an IP-Adapter (for attributes) and an InstantID-like module (for identity). This is a **compositional and incremental engineering step**, not a fundamental architectural invention. In their paper, the authors avoid using the standard term "Adapter" and instead coin the new term "Dual ReferenceNet"—to some extent, this obscures the limited architectural novelty of their method.

**Questions:**

Pls see weakness.

**Details Of Ethics Concerns:**

Portrait generation may include bias.

---

> ### Author Response · Authors · 2025-11-25
> **Response to Reviewer 5oxU (1/4)**
>
> **[W1] depends on LivePortrait’s self-reenactment ...may not be entirely fair.**
>
> Thank you for the helpful comment. We respectfully clarify that none of our quantitative evaluations rely on LivePortrait, and the comparisons in the main paper are entirely fair. The confusion likely stems from a distinction between our standard evaluation protocol and the practical inference trick described in the Appendix. We break down the difference below:
>
> - Quantitative Evaluation (Standard Protocol): For all quantitative results reported in our paper and tables, we follow the standard self-reenactment setting. The facial keypoint guidance is extracted directly from the Ground Truth (GT) videos, not generated by LivePortrait. Since both our method and the baselines utilize the same GT keypoints for evaluation, the comparison is strictly fair.
>
> - Practical Cross-Identity Inference (Appendix A.3): The procedure described in Appendix A.3 using LivePortrait is an optional and general preprocessing step for qualitative cross-identity demonstrations (where the driving identity differs from the source). In these real-world scenarios, significant shape/scale discrepancies between the source and driver can degrade performance for any method (including baselines). The LivePortrait can be used to align the driving keypoints to the source face's shape. This is an implementation detail for obtaining more aligned keypoint guidance input and is not part of our core model architecture.
>
> We appreciate the reviewer raising this concern and will revise the evaluation description to avoid any potential confusion.

---

> ### Author Response · Authors · 2025-11-25
> **Response to Reviewer 5oxU (2/4)**
>
> **[W2] metrics may be unreliable**
>
> Thank you for the thoughtful comment.
> To address this point, we briefly clarify the self-attribute transfer setting used in our evaluation.
>
> In an ideal evaluation, one would use groundtruth videos that contain before and after versions of the attribute transfer. Since such paired data does not exist, the commonly used alternative in attribute transfer and VTON literature is the self-attribute(or garment) transfer setting [1, 2]. In this setup, we take a groundtruth video and select two arbitrary frames as the portrait image and the attribute image. The generated video is then compared against the original ground truth video using reconstruction metrics such as L1, PSNR, SSIM, and LPIPS, as reported in Tab. 1 and Tab. 2. Although this setting cannot directly evaluate genuine cross-identity transfer, it still provides useful information because the attribute region in the portrait image and the identity region in the attribute image are **masked out** (see the inputs in Fig. 2). This forces the model to combine complementary cues in order to reconstruct the video.
>
> The ablation study shown in Tab. 2 is via self-attribute transfer setting.
> However, in the ``full ref. image input'' ablation where masks are not used, the model can simply copy either the portrait or the attribute input because both already contain the required face and attribute cues. As a result, the model can obtain strong reconstruction metrics in Tab. 2 without learning true disentanglement.
>
> Importantly, this shortcut is specific to this variant: for all other ablations, the identity region in the attribute image and the attribute region in the portrait image are masked out, preventing such leakage and ensuring a fair and meaningful comparison. The limitation of the unmasked shortcut becomes evident in cross-attribute transfer, where the model fails to separate identity and attribute cues when the two inputs come from different sources, as shown in Fig. 5. We therefore suggest interpreting Fig. 5 together with Tab. 2 to understand this contrast. The baselines in Tab. 1 are also free from this identity leakage issue, which means that the self-attribute transfer–based quantitative evaluation there remains fair and trustworthy despite the shortcut observed in the full-ref ablation of Tab. 2. Moreover, in Tab. 1, we also report cross-attribute transfer results using perceptual and video-quality metrics (mCLIP, mDINO, ID SIM, VFID) to directly assess cross-identity behavior, complementing the self-attribute transfer evaluation.
>
> [1] Choi, Seunghwan, et al. "Viton-hd: High-resolution virtual try-on via misalignment-aware normalization." Proceedings of the IEEE/CVF conference on computer vision and pattern recognition. 2021.
>
> [2] Morelli, Davide, et al. "Dress code: High-resolution multi-category virtual try-on." Proceedings of the IEEE/CVF conference on computer vision and pattern recognition. 2022.

---

> ### Author Response · Authors · 2025-11-25
> **Response to Reviewer 5oxU (3/4)**
>
> **[W3] Fairness of comparisons with two stage pipelines**
>
> We thank the reviewer for this insightful comment. We believe the reviewer’s observation regarding the inherent disadvantages of two-stage methods further supports the core motivation of our work. While the benefits of a single-stage approach are conceptually clear, realizing it is fundamentally challenging due to the absence of paired video-level training data for attribute transfer. In practice, paired data is typically available only at the image level, and most video-based methods therefore focus solely on the animation problem using generic in-the-wild videos. This has forced prior approaches to address either attribute transfer or animation in isolation, rather than both in a unified framework. Our key contribution is a self-reconstruction training strategy that enables end-to-end learning of both attribute transfer and animation directly from in-the-wild, uncurated videos, without requiring any dedicated attribute-transfer video pairs.
>
> Regarding the baselines, since there is currently no direct single-stage method capable of jointly addressing both attribute transfer and animation, we compare our approach against the strongest feasible two-stage pipelines composed of state-of-the-art image editing and video animation models. These combinations represent the practical upper bound of what can be achieved when the two problems are solved separately using existing tools. Demonstrating that our unified model outperforms such pipelines highlights the effectiveness and necessity of our single-stage formulation. If the reviewer is aware of alternative pipelines that provide a fairer comparison, we would be glad to include them in a revised version.

---

> ### Author Response · Authors · 2025-11-25
> **Response to Reviewer 5oxU (4/4)**
>
> **[W4] IP-Adapter vs. ReferenceNet**
>
> We appreciate the reviewer’s comment, but we respectfully argue that the comparison with IP-Adapter and InstantID misunderstands the architectural role of ReferenceNet in animation and video diffusion models.
>
> The ReferenceNet architecture, introduced in pioneering diffusion-based human animation frameworks such as AnimateDiff [Guo et al., 2023], Animate Anyone [Hu et al., 2024], and Champ [Zhu et al., 2024], is specifically designed to address identity propagation and fine-grained detail consistency across frames (Line 192–194 in our main paper). Its structure and role are fundamentally different from lightweight per-image conditioning modules such as IP-Adapter or InstantID, which operate at the image level and are not tailored for preserving detailed identity cues in long video sequences. As also discussed and empirically validated in Animate Anyone [Hu et al., 2024] (see the SD-I2V baseline in their Tables 1–3), IP-Adapter–style approaches fall short in maintaining fine-grained identity details compared to ReferenceNet-based designs. We kindly refer the reviewer to the related work for a more comprehensive discussion.
>
> To avoid confusion, we emphasize that ReferenceNet (and the term itself) is not introduced by our work, but is an established architectural component in prior character animation models. Our contribution lies in extending this design to a Dual-ReferenceNet tailored for the joint problem of video generation and facial attribute transfer. Importantly, this is not a trivial duplication of a single ReferenceNet. Rather, the dual design is a necessary structural component for our self-reconstruction training strategy on uncurated, in-the-wild videos: it enables the model to disentangle identity and attribute cues without identity leakage while learning attribute transfer from videos where no explicit paired supervision exists. As shown in our ablation studies, this dual formulation is critical for stable training and leads to clear improvements over the standard single ReferenceNet.
>
>
> **Ethics Concerns: Portrait Generation Bias**
>
> Thank you for raising the concern regarding potential bias in portrait generation.
>
> Our method does not perform portrait generation. Instead, it takes a given portrait and an attribute image as inputs and performs portrait animation with attribute transfer. Because our system relies entirely on the provided images rather than synthesizing new identities, the usual sources of bias associated with generative portrait models do not arise in our setting.
> The resulting animation videos, as shown in the supplemental material, demonstrate high-quality identity preservation and accurate attribute transfer. These results show that the method faithfully follows the input portrait without altering demographic attributes or introducing unintended bias.
>
> We hope this clarifies why concerns related to portrait-generation bias are not applicable to our approach.

---

### Author Response · Authors · 2025-12-02
**General Response**

We sincerely thank all reviewers (5oxU, 2dwE, VRMj, 3NqY) for their careful reading of our submission and for
providing constructive and insightful feedback. We are also grateful to the AC and SAC for coordinating the overall
review process. We deeply appreciate the time and effort each of you has invested in evaluating our work.

We are encouraged by the reviewers’ recognition of the strengths of our submission, and we value the thoughtful
comments that will help us further improve the paper.

* **Clever self-reconstruction strategy**: ".. effectively eliminates the dependence on paired attribute data .. making it
an important advancement in attribute transfer." (3NqY), ".. does not need paired data" (5oxU), ".. eliminating the
need for expensive paired data .." (VRMj).

* **High technical originality**: "dual-reference architecture .. a novel design in diffusion-based frameworks." (3NqY),
".. design outperforms single ReferenceNet variants" (VRMj).

* **High quality results**: ".. significantly faithful outcomes on attribute transfer with high quality and relatively no
visual artifacts." (2dwE), "mask expansion and augmentation significantly improves robustness under cross-identity
attribute generation, ensuring consistent identity preservation." (3NqY), ".. explains its strong cross-identity results"
(VRMj).

* **Emergent properties**: "multi-attribute composition and smooth interpolation in a single forward pass, indicating
strong representational power and generalization." (3NqY), ".. the concat-based module produces pretty plausible
results." (2dwE), ".. enables multi-attribute composition and attribute interpolation .." (VRMj)

**Manuscript Revision Summary**

We thank the reviewers for their constructive feedback and have revised the manuscript and supplementary material
accordingly.

The key updates are summarized below (with changes marked in **blue**):

1. **Automatic Prompt Generation for Attribute-Aware Mask Expansion (3NqY)**:
We added a detailed explanation of how prompts are generated in a fully automatic manner within the attribute-aware mask expansion module (main manuscript, Lines 287–289).
2. **Clarification on Quantitative Comparison and Keypoint Guidance Generation (5oxU)**:
We explicitly clarified that LivePortrait is not used in any part of the quantitative comparison for keypoint guidance generation (supplementary material, Lines 808–809).
3. **Sensitivity Analysis with respect to Attribute Masks (3NqY, 2dwE)**:
We included a sensitivity analysis demonstrating how different attribute masks influence performance, together with the corresponding visualization in Figure 9 (supplementary material, Lines 843–849).
4. **Generalize to the Cartoon Domain (3NqY)**:
We present the results of applying our method to the cartoon domain without any additional fine-tuning or optimization (supplementary material, Lines 1001–1005, Figure 15).
5. **Failure Case Analysis (3NqY, VRMj)**:
We introduced additional failure case discussions covering two challenging conditions: conflicting lighting and occlusion, accompanied by Figures 16 and 17 (supplementary material, Lines 1008–1025).
6. **Additional Explanation for the Self-Attribute Transfer Evaluation (5oxU)**:
We supplemented the evaluation protocol and interpretation for the self-attribute transfer scenario, providing more clarity on how the setting is constructed and assessed (supplementary material, Lines 1070–1079).
7. **Further Details on the Full Ref. Image Input Variant (Ablation Study) (5oxU, VRMj)**:
We provided additional explanation and motivation for the "Full Ref. Image Input" variant in the ablation study (supplementary material, Lines 1101–1111).
8. **Addition of Supplemental Video (3224_rebuttal.mp4)**:
We added a supplemental video file (3224_rebuttal.mp4) during the rebuttal process to address the raised concerns.
The video includes demonstrations of:
    * generalization to non-human domains (3NqY),
    * mask sensitivity (2dwE, 3NqY),
    * effect of the number of attributes (2dwE),
    * failure cases (VRMj, 3NqY).

We hope these revisions address all concerns and further clarify the contributions of our work.

---

### Meta-Review · Area_Chair_BnPM · 2025-12-12

**Summary:**

The paper introduces a dual-reference diffusion model for portrait animation with attribute transfer, trained via a self-reconstruction scheme on unpaired videos. Two reviewers (2dwE, 3NqY) are positive about the idea, the visual quality, and the training strategy, while two reviewers (5oxU, VRMj) raise concerns about novelty, reliance on external tools, and limited generality.

After reading the paper and the discussion, I agree that the core architectural novelty is somewhat incremental, and the pipeline relies on several strong external components. On the other hand, the paper is technically competent, the results are consistently strong, and the authors did a solid job addressing key fairness and evaluation questions during rebuttal. The dual-reference design combined with the self-reconstruction training setup seems to provide a practical and effective way to learn attribute transfer without paired data, which is a real hurdle in this space. Overall, despite remaining reservations about how “new” the architecture is in isolation, I lean toward an accept recommendation because the work presents a coherent system, clear empirical value, and a reasonably convincing set of ablations and analyses.

**Reviewer Concerns:**

Concerns that are largely addressed:
1. The authors clarified that LivePortrait is not used in quantitative evaluation, so fairness issues in the core metrics are mostly addressed.

2. They gave a reasonable explanation of why the “full ref. image input” variant can game self-reconstruction metrics but fails in cross-identity transfer.

3. They added mask sensitivity analysis and more failure examples, which partly respond to 2dwE and 3NqY.

4. Several factual errors in VRMj’s review (for example about video duration and nonexistent modules) were correctly pointed out and should be discounted.

Concerns that remain substantial:

1. I agree with 5oxU that the architectural contribution is closer to a well-executed composition than a fundamentally new architectural concept. Even with the clarification about ReferenceNet versus IP-Adapter style conditioning, the dual-branch design can read as an incremental step.

2. The method depends on external tools (segmentation, outpainting, keypoint guidance). This reduces how self-contained the contribution feels, and the paper could do a better job quantifying how sensitive final results are to each external block.

3. The effective attribute space is limited by what the segmentation masks can reliably capture. When masks are imperfect, outputs degrade, and in practice some manual correction may be needed. This also partially explains why only a small set of attributes is demonstrated.

4. VRMj’s more speculative points about very long videos or very difficult inputs are factual errors, but the underlying concerns on reliance on external components and limited attribute coverage remain valid.

I also found the qualitative comparison in Fig. 4 is not the cleanest example for isolating the benefit of the dual-reference design, since masking itself can explain much of the visual advantage in that case. I would encourage the authors to add a more balanced example or explicitly separate “mask benefit” versus “architecture benefit” in the narrative.

Overall, the rebuttal improves clarity and strengthens confidence in the experimental protocol. While some novelty and dependency concerns remain, they do not outweigh the demonstrated effectiveness and the practical value of the approach.

**Reviewer Scores:**

Reviewer 5oxU (2, reject): Their main concern is incremental novelty. I expect they would keep the score at 2.

Reviewer 2dwE (8, accept): Strongly positive on quality and practicality. The rebuttal addressed their questions, so they would likely remain at 8.

Reviewer VRMj (4, borderline reject): Some criticisms are based on factual errors, but the broader concerns on external dependencies and limited attribute coverage remain. They would likely stay at 4.

Reviewer 3NqY (8, accept): Very positive on originality and execution. Their concerns were largely addressed, so they would likely remain at 8.

Given that the rebuttal resolved several key issues, my overall recommendation is accept.

---

### Decision · Program_Chairs · 2026-01-26

Accept (Poster)